# Plakoglobin transmits tension across VE-cadherin for vascular leak formation and leukocyte diapedesis

Neha Uttekar [ID], Annette Artz, Vallari Ghanekar [ID], Pragya Kaul [ID], Jessica Heinrichs, Rebekka I Stegmeyer, Gizem Gülevin Takir [ID], Astrid F Nottebaum [ID] & Dietmar Vestweber [ID] ✉

## Abstract

VE-cadherin controls endothelial junction integrity, and thereby inflammation-induced vascular permeability and leukocyte extravasation. The adhesive function of VE-cadherin is influenced by its binding to β-catenin, which is linked by α-catenin to actin. Plakoglobin can replace β-catenin in such complexes, and both types of complexes co-exist in endothelial cells. Here, we have investigated whether β-catenin and plakoglobin differ in their relevance for controlling endothelial junctions. Based on gene silencing in vitro and conditional endothelium-specific gene inactivation in mice in vivo, we found that both leukocyte diapedesis through endothelium and induction of vascular permeability by inflammatory mediators depend on plakoglobin, but not β-catenin. Mechanistically, we demonstrated that plakoglobin is crucial for the generation of tension across VE-cadherin by transmigrating leukocytes and by inflammatory mediators, whereas β-catenin was dispensable in this context. Transgenic mice expressing a VE-cadherin tension sensor revealed that plakoglobin is essential in vivo for histamine-induced tension across VE-cadherin. Thus, plakoglobin, but not β-catenin, is needed for leukocyte diapedesis, the induction of vascular permeability, and the stimulation of mechanical tension across VE-cadherin.

**Keywords** Endothelial Junctions; Vascular Permeability; Leukocyte Trafficking
**Subject Categories** Cell Adhesion, Polarity & Cytoskeleton; Vascular Biology & Angiogenesis

## Introduction

Endothelial cells form tight and adherens junctions which enables them to function as a barrier that controls leukocyte extravasation and leakage of plasma solutes. The integrity of this barrier is challenged by inflammatory processes which on the one hand allows leukocytes to assess sites of infection. Yet, on the other hand, in extreme cases, leakage can also have dire consequences when edema formation becomes systemic and leads to multiple organ failure. A major constituent of endothelial adherens junctions and a target for processes that interfere with junction integrity is the homophilic adhesion molecule VE-cadherin.

The adhesive function of cadherins is dependent and influenced by their anchoring to the actin cytoskeleton. This link strongly supports cadherin-mediated cell adhesion and allows actin-remodeling and actomyosin driven mechanical forces to modulate cell adhesion. The link to actin is mediated by the catenins (Ozawa et al, 1989), with β-catenin binding to the C-terminus of cadherins and with α-catenin binding to β-catenin thereby serving as a linker to actin (Huber and Weis, 2001; Pokutta and Weis, 2000). β-catenin can be replaced by plakoglobin, a protein also known as γ-catenin (Ozawa et al, 1989; Butz et al, 1992; Butz and Kemler, 1994).

Plakoglobin is a structural homolog of β-catenin (McCrea et al, 1991; Butz et al, 1992; Peifer et al, 1992). Both contain a large central area (with 83% sequence similarity) harboring 13 armadillo repeats with which each protein binds to the C-terminus of cadherins thereby linking them to α-catenin (Aktary et al, 2017, review). It is not known whether plakoglobin and β-catenin are of equivalent relevance as linker in such complexes. Neither is it known why there are two such linkers. The relative amounts of these two types of complexes can strongly vary as was shown for different epithelial cell lines (Butz and Kemler, 1994).

Besides this adapter function, β-catenin is well known as a central target in wnt induced transcriptional regulation, where β-catenin acts as a co-factor of the transcription factor TCF/LEF (Clevers and Nuse, 2012). In contrast, plakoglobin binds much less efficient to TCF/LEF. Yet, it is well documented that it can influence transcription by β-catenin-dependent and independent ways (Aktary et al, 2017; Zhurinsky et al, 2000).

Plakoglobin was originally identified as a desmosomal protein which later was also localized at adherens junctions (Cowin et al, 1986). In contrast to plakoglobin, β-catenin is normally not found in desmosomal plaques (Choi et al, 2009). At desmosomes, plakoglobin binds to desmosomal cadherins and to desmoplakin which links it to intermediate filaments (Schmidt et al, 1994). Since endothelial cells, in contrast to epithelial cells and cardiomyocytes, are devoid of desmosomes, staining for desmoplakin at endothelial junctions raised the idea that VE-cadherin might recruit desmoplakin via plakoglobin to adherens junctions, which would open the possibility of vimentin recruitment (Valiron et al, 1996; Kowalczyk et al, 1998). However, no co-immunoprecipitations of VE-cadherin with desmoplakin from endothelial cells are documented and today

Max Planck Institute for Molecular Biomedicine, Münster, Germany. ✉E-mail: vestweb@mpi-muenster.mpg.de

we know that endothelial cells express the desmosomal cadherin desmoglein2 (DSG2) at endothelial cell contacts, despite the lack of desmosomes (Ebert et al, 2016). Thus, DSG2 could be an alternative explanation for the recruitment of desmoplakin to endothelial cell contacts.

The first hint that VE-cadherin-catenin complexes containing either β-catenin or plakoglobin would behave differently with respect to the regulation and support of junctions goes back to the observation that recently confluent endothelial cells contain only β-catenin but no plakoglobin at cell contacts, whereas plakoglobin is found much later when junctions are fully mature (Lampugnani et al, 1995). This was confirmed by others showing that recovery of junctions after Ca²⁺-depletion first coincides with β-catenin recruitment which is then followed by plakoglobin (Schnittler et al, 1997). However, whether β-catenin or plakoglobin differ in their way they support the influence of the actin cytoskeleton on VE-cadherin function and junction integrity is not known.

Here, we have directly compared the relevance of β-catenin and plakoglobin for the control of endothelial junctions under inflammatory conditions in vitro and in vivo. For this we generated conditional gene inactivated mice lacking either β-catenin (in an inducible way) or plakoglobin selectively in endothelial cells. We found that only plakoglobin, but not β-catenin is in vitro and in vivo required for the induction of vascular leaks and for leukocyte extravasation by inflammatory mediators. Searching for differences between the functions of β-catenin and plakoglobin in the respective VE-cadherin-catenin complexes, we used a Förster resonance energy transfer (FRET)-based VE-cadherin tension sensor. We could show in vitro that transmigrating leukocytes as well as vascular permeability inducing mediators required plako-globin for generating tension across VE-cadherin, whereas β-catenin was dispensable. Generating and analyzing knock in mice expressing the same VE-cadherin tension sensor instead of endogenous VE-cadherin allowed us to show that histamine triggers tension across VE-cadherin in vivo and just like in vitro requires plakoglobin for this effect. Thus, plakoglobin and not β-catenin is in vitro and in vivo selectively required for the interference with endothelial junction integrity in inflammation and is needed for tension induction across VE-cadherin during these processes.

## Results

### Plakoglobin but not β-catenin supports leukocyte diapedesis in vitro and in vivo

VE-cadherin is of central importance for the control of endothelial junctions and therefore for the regulation of leukocyte diapedesis. The link of VE-cadherin to actin is essential for junction integrity and this link can be mediated by either the β-catenin-α-catenin or the plakoglobin-α-catenin complex. Since it is not known whether these two complexes differ in their relevance for the control of endothelial junctions, we decided to test the influence of gene silencing of either β-catenin or plakoglobin on neutrophil transmigration through cultured endothelial cells. To this end, we treated HUVEC for 72 h with either control siRNA or siRNAs for β-catenin or plakoglobin. As shown by immunoblotting (Fig. 1A,B), β-catenin expression was reduced by 81% and plakoglobin by 86%.

Remarkably, in each case the expression of the other catenin was upregulated, leading to full compensation (Fig. 1B). To test the effect of gene silencing on leukocyte transmigration, we incubated siRNA treated HUVEC after 4 h TNF-α stimulation with human neutrophils under flow and quantified transmigrated and adherent neutrophils. As shown in Fig. 1C, plakoglobin silencing inhibited transmigration by 75% whereas no inhibitory effect was seen when HUVEC were pre-treated with β-catenin siRNA. No significant inhibitory effect was seen for the number of adherent neutrophils (Fig. 1D). Thus, plakoglobin but not β-catenin was required for proper neutrophil diapedesis in vitro.

To verify these results in vivo, we generated mice gene-inactivated in endothelial cells for either β-catenin (Ctnnb1) or plakoglobin (Jup). To this end, Jup^lox/lox mice were bred to Tek-Cre mice, which resulted in viable and healthy endothelial cell knock-out (ECKO) mice (Jup^ECKO). Since deletion of β-catenin in endothelium is embryonic lethal (Cattelino et al, 2003; Liebner et al, 2004) we bred Ctnnb1^lox/lox mice to tamoxifen-inducible Pdgfb-iCre transgenics, resulting in inducible endothelial knock-out (iECKO) mice (Ctnnb1^iECKO). Since even tamoxifen-induced full gene inactivation of β-catenin in endothelium is lethal due to the loss of claudins which leads to blood–brain barrier brake-down (Tran et al, 2016), we could not completely delete β-catenin in endothelium of our mice. As shown by immunoblotting of VE-cadherin immunoprecipitates from lung lysates, expression of plakoglobin was very efficiently inhibited in Jup^ECKO and β-catenin was partially inhibited in Ctnnb1^iECKO (Fig. 2A,B). Expression of the respective catenin counterpart was upregulated and fully compensated the loss of the other catenin (Fig. 2C,D). In line with this compensating effect, expression levels of VE-cadherin at endothelial junctions were not altered in Jup^ECKO or Ctnnb1^iECKO when compared to Jup^lox/lox and Ctnnb1^lox/lox mice as judged by whole-mount staining of their cremaster muscle (Fig. EV1). To test potential effects of plakoglobin and β-catenin gene inactivation on the extravasation of neutrophils in vivo, we analyzed neutrophil extravasation in the cremaster after 4 h intrascrotal stimulation with IL-1β by intravital microscopy. As shown in Fig. 2E–G neutrophil transmigration was reduced in Jup^ECKO mice by 43% when compared to Jup^lox/lox mice whereas rolling flux fraction and adhesion were not significantly altered. In contrast, neutrophil extravasation was not altered in Ctnnb1^iECKO when compared to Ctnnb1^lox/lox mice (Fig. 2H–J). Thus, as in vitro, neutrophil diapedesis in vivo depends on the presence of plakoglobin, whereas β-catenin is not required.

### Inflammatory mediators impair endothelial barrier integrity by a mechanism that depends on plakoglobin, but not β-catenin

Endothelial junctions control vascular permeability which is induced by various inflammatory mediators. Again, VE-cadherin is a major target in this process. Therefore, we tested whether plakoglobin and β-catenin are involved in inflammation-induced alteration of endothelial barrier integrity. As a parameter for barrier integrity, we analyzed electrical resistance to alternating current (ECIS) of HUVEC monolayers grown on fibronectin-coated electrode arrays. HUVEC were pretreated with either control, β-catenin or plakoglobin siRNA. Upon stimulation with thrombin, resistance dropped over time with a maximal decrease at about

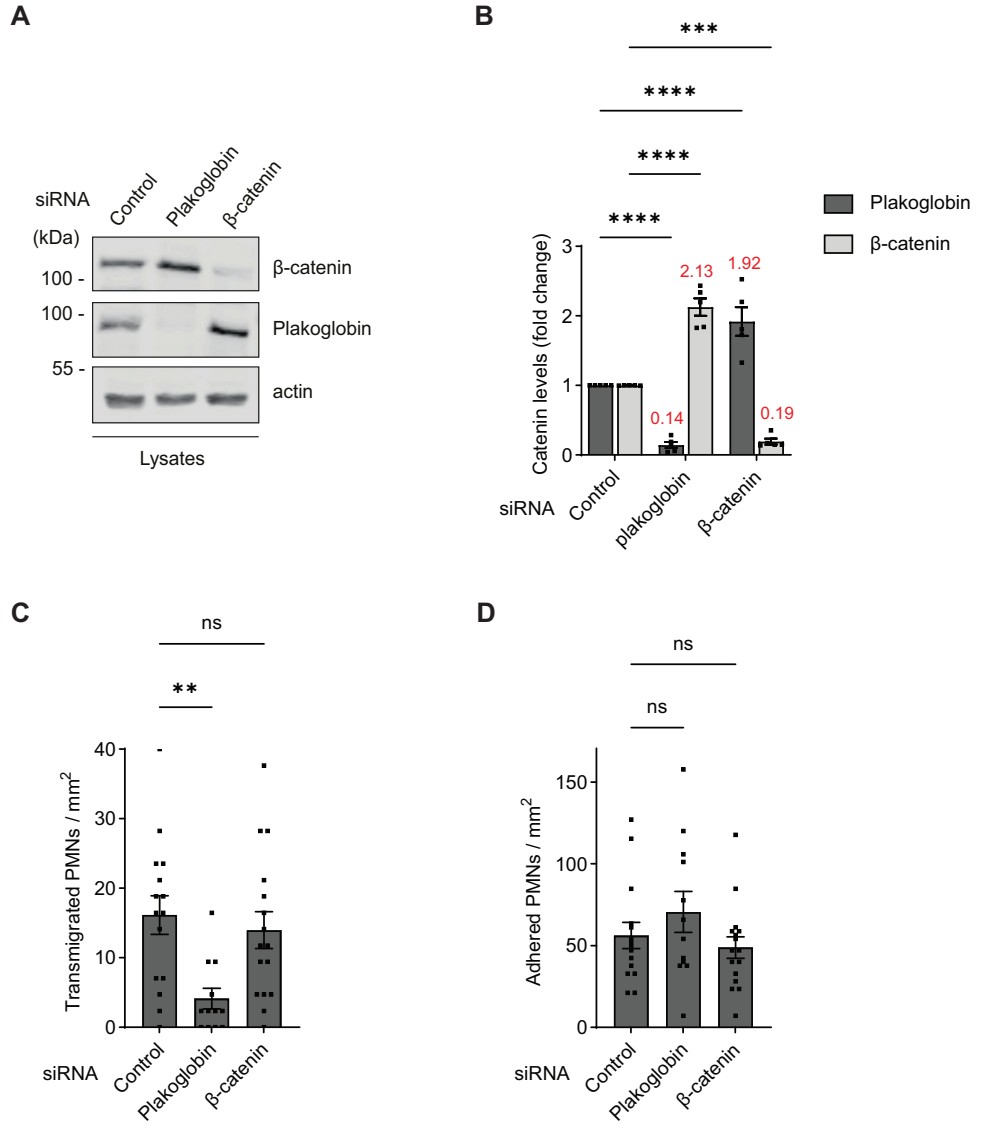

**Figure 1. Leukocyte transmigration is inhibited by the loss of plakoglobin but not β-catenin in vitro.**

(**A**) Representative immunoblot of lysates from HUVEC pre-treated with control, plakoglobin or β-catenin siRNA for 72 h, detected with antibodies against plakoglobin, β-catenin and actin. (**B**) Quantification of plakoglobin and β-catenin blot signals (from **A**) normalized to the amount of actin. Mean values for up- and down-regulation for each group are indicated in red. (**C, D**) Human PMNs (polymorphonuclear neutrophils) transmigrated through (**C**) and adhered to (**D**) 4 h TNFα stimulated HUVEC pre-treated with control, plakoglobin or β-catenin siRNA under flow. Data information: Data are mean ± SEM of five independent experiments in (**B**) or mean ± SEM of 15 (control), 12 (plakoglobin) and 16 (β-catenin) videos from three independent experiments in (**C, D**). ***$P = 0.0001$, ****$P < 0.0001$, two-way ANOVA (**B**), **$P = 0.006$, one-way ANOVA (**C, D**). ns, not significant. Source data are available online for this figure.

30 min across control siRNA treated cells (Fig. 3A). Drop and recovery were largely similar for β-catenin siRNA treated cells (Fig. 3B), whereas plakoglobin siRNA treated cells showed only a small decrease in resistance (Fig. 3C). Quantitation of the drop in resistance at 30 min after thrombin stimulation revealed no significant difference between control and β-catenin siRNA treated cells whereas for plakoglobin siRNA treated cells the drop in resistance was reduced by 60% (Fig. 3D). We conclude that the mechanism whereby thrombin impairs electrical resistance across endothelial monolayer depends on plakoglobin but not on β-catenin.

To investigate the relevance of β-catenin and plakoglobin for the induction of vascular permeability by inflammatory mediators in vivo, we analyzed histamine-induced leak formation in the cremaster of our plakoglobin and β-catenin conditional KO mice described above. Leakage was detected by intravenous injection of Evans blue dye together with histamine for 15 min followed by removal of cremaster tissue and quantification of albumin-adsorbed dye extracted from it. As shown in Fig. 4A, induction of vascular permeability in $Jup^{ECKO}$ mice was 38% less than in $Jup^{lox/lox}$ mice. In contrast, no significant change in permeability induction was observed when $Ctnnb1^{iECKO}$ and $Ctnnb1^{lox/lox}$ mice were

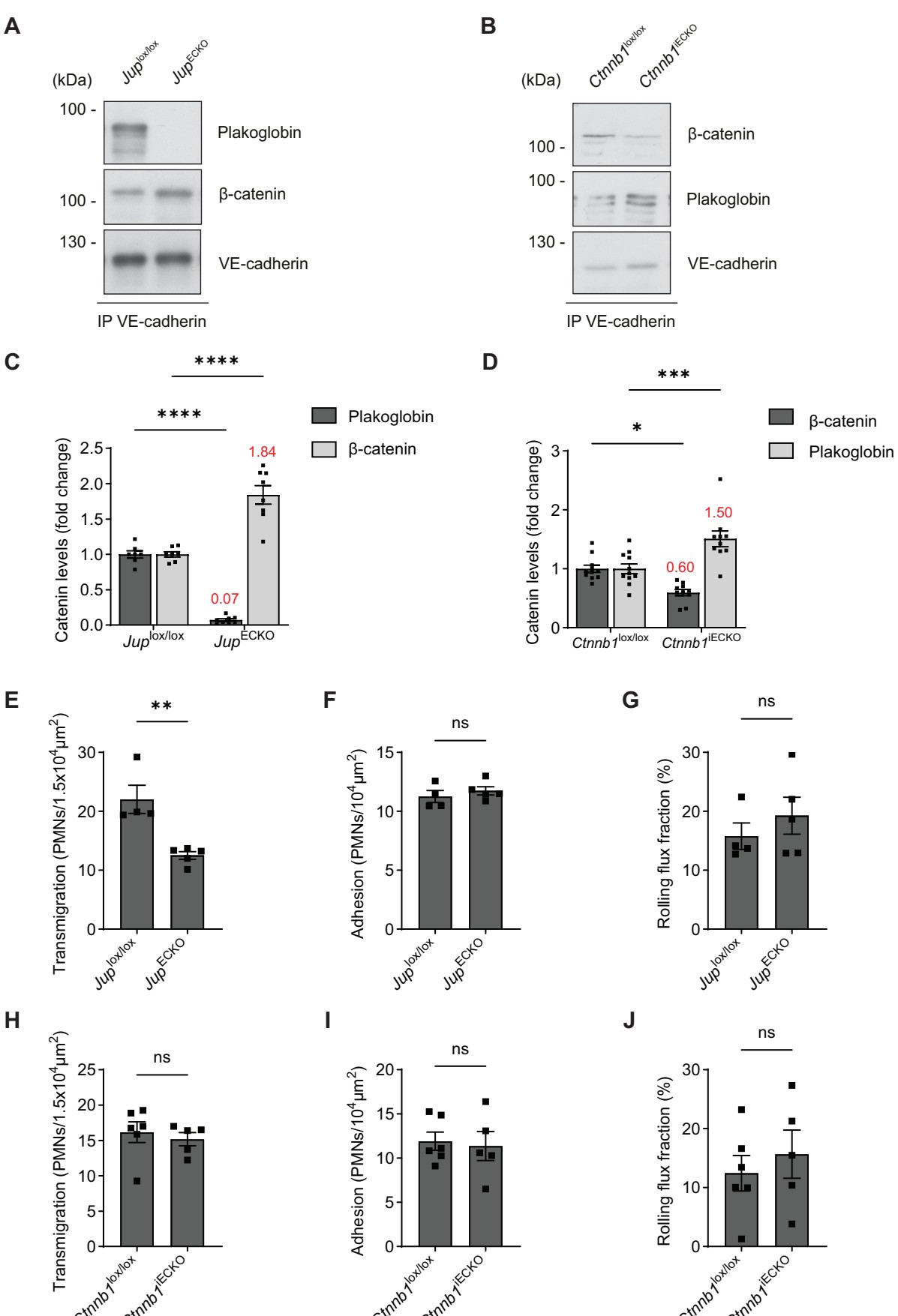

◀ **Figure 2.  Neutrophil extravasation is reduced by endothelial specific gene inactivation of plakoglobin but not β-catenin in vivo.**

(A, B) Representative immunoblot of VE-cadherin immunoprecipitates from lung lysates of *Jup*^lox/lox and *Jup*^ECKO mice (A) or *Ctnnb1*^lox/lox and *Ctnnb1*^iECKO mice (B), detected with antibodies against plakoglobin, β-catenin and VE-cadherin. (C, D) Quantification of plakoglobin and β-catenin blot signals (from A, B) normalized to amount of VE-cadherin. Mean values for up- and down-regulation for each group are indicated in red. (E–J) Transmigration (E, H), adhesion (F, I) and rolling flux fraction (G, J) of neutrophils in cremaster tissue from *Jup*^lox/lox and *Jup*^ECKO mice (E–G) or *Ctnnb1*^lox/lox and *Ctnnb1*^iECKO mice (H–J), given intrascrotal injection of IL-1β (50 ng) 4 h before intravital microscopy. Data information: Mean ± SEM of at least 7 mice per group (C), or at least 10 mice per group (D) or at least 27 vessels from four to six mice per group (E–J). *$P = 0.0109$, ***$P = 0.001$, ****$P < 0.0001$, two-way ANOVA (C, D), **$P = 0.0037$, Unpaired t-test (E–J). ns, not significant. Source data are available online for this figure.

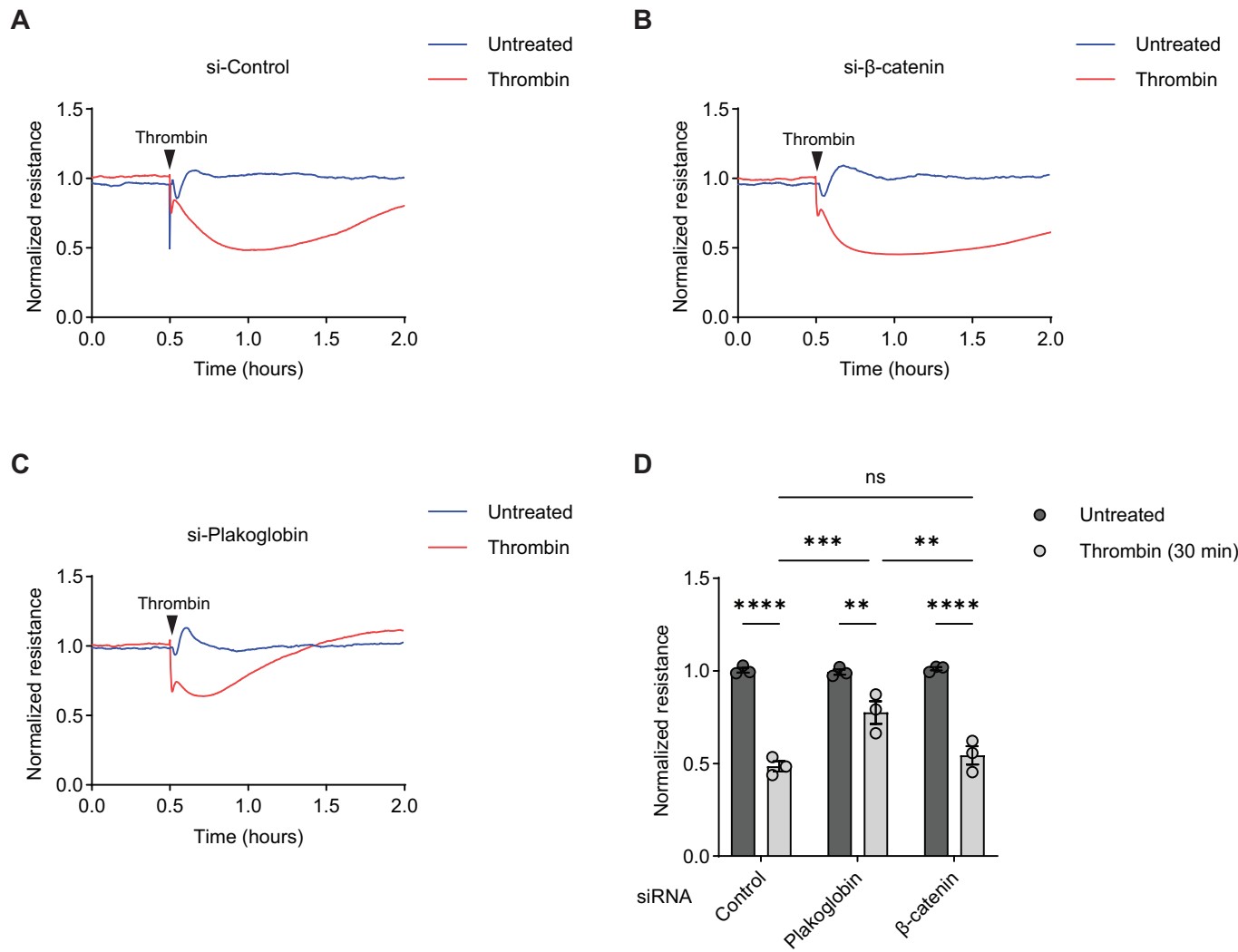

**Figure 3.  Thrombin-mediated impairment of endothelial barrier integrity in vitro depends on plakoglobin, but not on β-catenin.**

(A–C) HUVEC were transfected with control (A), β-catenin (B) or plakoglobin (C) siRNA and grown to confluency on fibronectin-coated electrode arrays. Cells were treated with 1 U/mL thrombin (red line) or left untreated (blue line), and the electrical resistance was monitored in time by ECIS. (D) Quantification of results in (A–C), presented as HUVEC monolayer resistance after 30 min of thrombin treatment, normalized to the resistance of untreated HUVEC monolayer. Data information: Data are representative of three experiments (A–D) (Mean ± SEM in D). **$P = 0.0084$ (si-Plakoglobin:untreated vs. si-Plakoglobin:thrombin), **$P = 0.0053$ (si-Plakoglobin:-thrombin vs. si-β-catenin:thrombin), ***$P = 0.0008$, ****$P < 0.0001$, two-way ANOVA (D). ns, not significant. Source data are available online for this figure.

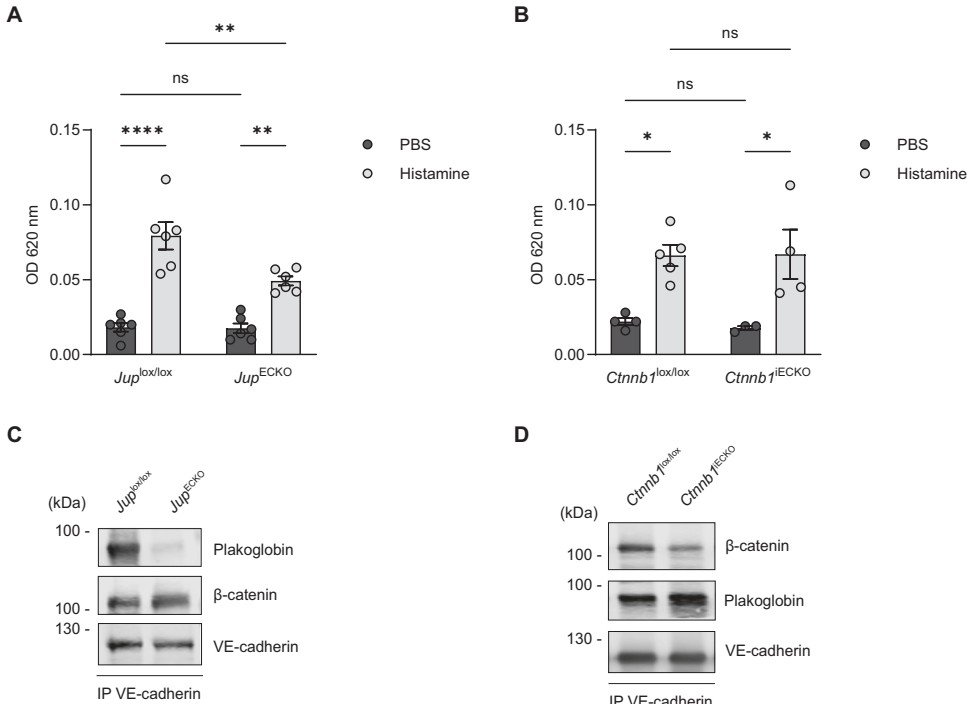

**Figure 4. Plakoglobin, but not β-catenin, is important for vascular permeability induction by histamine in the mouse cremaster muscle.**

(A, B) Vascular permeability in cremaster tissue of $Jup^{lox/lox}$ and $Jup^{ECKO}$ mice (A) or $Ctnnb1^{lox/lox}$ and $Ctnnb1^{iECKO}$ mice (B), given intravenous injection of Evans blue dye together with histamine for 15 min, followed by removal of cremaster tissue and quantification of dye extracted from it (OD 620 nm). (C, D) Representative immunoblot of VE-cadherin immunoprecipitates from lung lysates of $Jup^{lox/lox}$ and $Jup^{ECKO}$ mice (C) or $Ctnnb1^{lox/lox}$ and $Ctnnb1^{iECKO}$ mice (D), detected with antibodies against plakoglobin, β-catenin and VE-cadherin. Data information: Mean ± SEM of six mice per group (A) or three to five mice per group (B). $*P = 0.0217$ ($Ctnnb1^{lox/lox}$:PBS vs. $Ctnnb1^{lox/lox}$:Histamine), $*P = 0.0244$ ($Ctnnb1^{iECKO}$:PBS vs. $Ctnnb1^{iECKO}$:Histamine), $**P = 0.0034$ ($Jup^{lox/lox}$:Histamine vs. $Jup^{ECKO}$:Histamine), $**P = 0.0022$ ($Jup^{ECKO}$:PBS vs. $Jup^{ECKO}$:Histamine), $****P < 0.0001$, two-way ANOVA (A, B). ns, not significant. Source data are available online for this figure.

compared (Fig. 4B). The efficiency of gene inactivation was determined by immunoblotting of VE-cadherin immunoprecipitates of lung lysates for the respective catenin (Fig. 4C,D). Again, we conclude that similar to our in vitro results, it is plakoglobin but not β-catenin which is needed for vascular permeability induction by inflammatory mediators in vivo.

## Overexpression of β-catenin does not interfere with neutrophil diapedesis and thrombin-induced loss of junction integrity

Since plakoglobin silencing is compensated by the upregulation of β-catenin expression, we could not exclude that it is the upregulation of β-catenin instead of the loss of plakoglobin which causes the effects we observed for endothelial junctions. To clarify this, we overexpressed β-catenin by adenovirus transduction and treated control cells with adenovirus-LacZ vector. Efficiency of β-catenin expression was tested by immunofluorescence staining in cells that had been pre-treated with β-catenin siRNA (Fig. 5A). Overexpression of β-catenin in HUVEC which were not pre-treated with siRNA led to an almost two-fold increase in expression as tested by immunoblotting (Fig. 5B,C). Performing neutrophil transmigration assays under flow revealed that transmigration as well as attachment to endothelial cells were indistinguishable between HUVEC transduced with β-catenin or with LacZ

(Fig. 5D,E). Likewise, we found that thrombin-induced reduction of electrical resistance across HUVEC monolayers was largely indistinguishable between cells transduced with β-catenin or LacZ (Fig. 5F–H). We conclude, that it is the lack of plakoglobin, and not the compensating upregulation of β-catenin which impairs neutrophil diapedesis and thrombin-induced break-down of junction integrity.

In contrast to the ECIS measurements shown in Fig. 3, where cells had not been treated with adenovirus vectors, recovery of electrical resistance was accelerated. When we compared such measurements directly with those of cells not treated with adenovirus, we could verify that it is indeed the adenovirus treatment which accelerates this process (Fig. EV2). Our conclusions are not affected by this phenomenon since we compared β-catenin and LacZ transduced cells in our transmigration and ECIS assays.

## Plakoglobin and β-catenin do not differ in their distribution at junctions nor do they differentially affect VE-cadherin mobility

In order to find out why plakoglobin and β-catenin play different roles in processes which interfere with junction integrity, we tested whether the mobility of VE-cadherin is differentially affected by silencing either plakoglobin or β-catenin. HUVEC were transduced

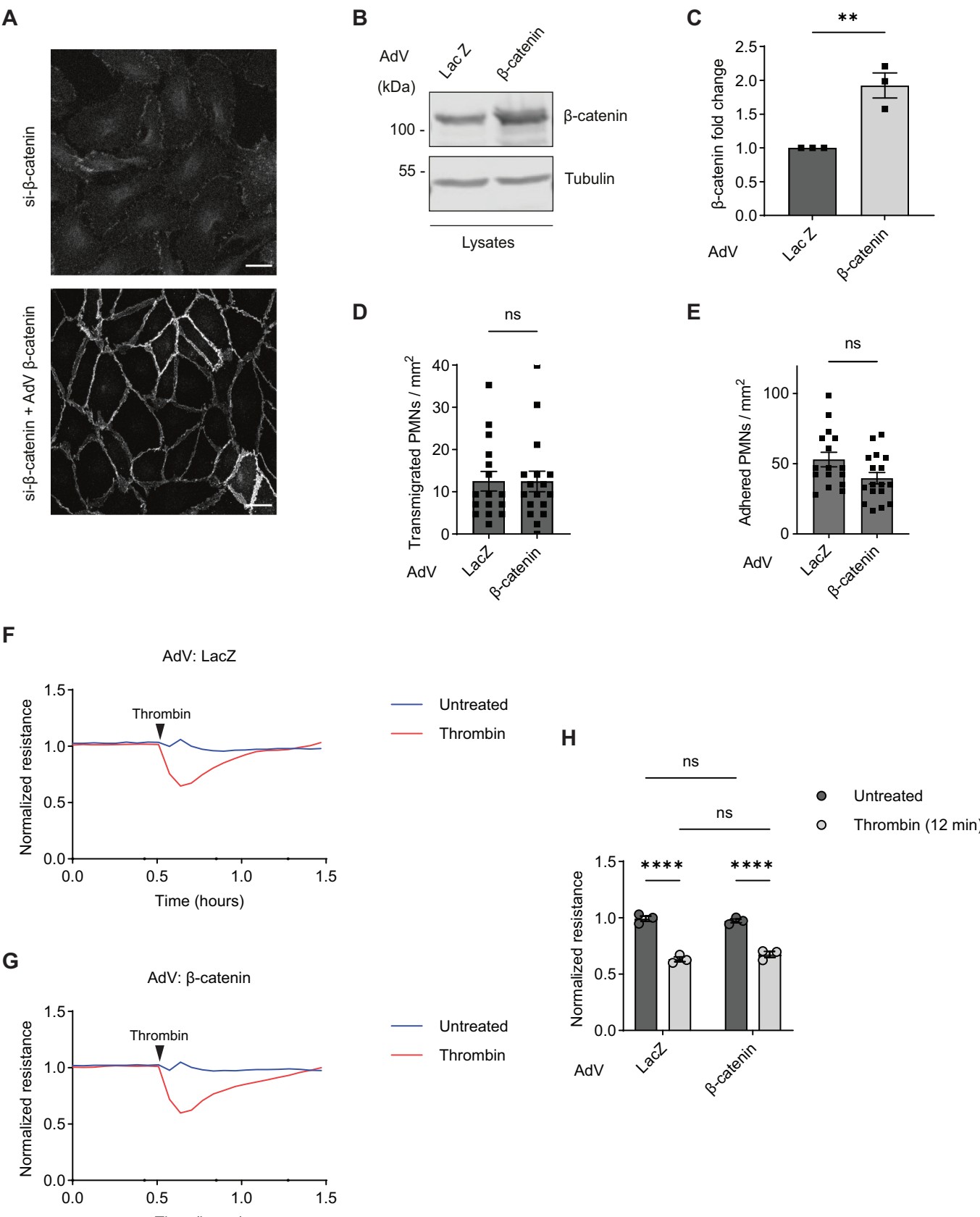

◄ **Figure 5.  β-catenin overexpression does not affect leukocyte transmigration and thrombin-mediated impairment of endothelial barrier integrity.**

(**A**) Immunofluorescence staining for β-catenin of HUVEC transfected with β-catenin-specific siRNA (si-β-catenin) and transduced with β-catenin adenovirus. (**B**) Representative immunoblot of lysates from HUVEC transduced with LacZ or β-catenin adenovirus for 48 h, analyzed with antibodies against β-catenin and tubulin. (**C**) Quantification of β-catenin blot signal (from **B**) normalized to the amount of tubulin. (**D, E**) Human PMNs transmigrated through (**D**) and adhered to (**E**) 4 h TNFα stimulated HUVEC transduced with LacZ or β-catenin adenovirus under flow. (**F, G**) HUVEC were transduced with LacZ (**F**) or β-catenin (**G**) adenovirus and grown to confluency on fibronectin-coated electrode arrays. Cells were treated with 1 U/mL thrombin (red line) or left untreated (blue line), and the electrical resistance was monitored in time by ECIS. Representative graph shows resistance after normalization to the resistance of untreated HUVEC monolayer. (**H**) Quantification of results in (**F, G**), presented as HUVEC monolayer normalized resistance after 12 min of thrombin treatment. Data information: Data are mean ± SEM of three independent experiments in (**C**), mean ± SEM of 16 (LacZ) and 17 (β-catenin) videos pooled from three independent experiments in (**D, E**) or are representative of three experiments (**F–H**) (Mean ± SEM in **H**). **$P = 0.0076$, unpaired t-test (**C**), Mann–Whitney test (**D, E**), ****$P < 0.0001$, two-way ANOVA (**H**). ns, not significant. Source data are available online for this figure.

with a VE-cadherin-fusion protein containing the chromophore YPet (VEC-TS-YPet, see below) and either treated with plakoglobin or β-catenin siRNA, followed by fluorescence recovery after photobleaching (FRAP) experiments. As shown in Fig. 6A, the time course of FRAP was nearly identical for cells no matter with which type of siRNA they were treated. Quantification revealed that the mobile fraction of VE-cadherin and the half-time recovery were indistinguishable between plakoglobin and β-catenin siRNA-treated cells (Fig. 6B,C). Thus, the mobility of VE-cadherin at endothelial junctions is independent of the type of catenin with which it associates.

Next, we tested whether the distribution of the two catenins at junctions of HUVEC would differ and whether the distribution pattern would be differently affected by stimulating HUVEC with either histamine or thrombin. As shown in Fig. EV3, indirect immunofluorescence of VE-cadherin and each of the two catenins revealed no significant difference in the distribution pattern of these three proteins as judged by images at high resolution and by images that allowed a broader overview over a larger number of cells. Neither thrombin nor histamine had differential effects on the distribution pattern of any of the three junctional proteins (Fig. EV3).

## Endothelial plakoglobin, but not β-catenin is needed for leukocyte induced tension across VE-cadherin during diapedesis

The dramatic difference of the relevance of plakoglobin and β-catenin for the control of leukocyte extravasation and vascular permeability induction raised the question whether plakoglobin and β-catenin differ in the way they function in the VE-cadherin-catenin complex. We have shown recently that leukocyte diapedesis triggers myosin activation in endothelial cells which leads to an increase of tension across VE-cadherin (Arif et al, 2021). Therefore, we decided to test whether plakoglobin and β-catenin would be of similar relevance for this effect. Our previous results about leukocyte induced tension across VE-cadherin were obtained by FRET measurements with the help of a VE-cadherin tension sensor (VEC-TS) that contained a force sensitive Förster resonance energy transfer (FRET) module inserted into the cytoplasmic tail of VE-cadherin between the p120 and the β-catenin binding site (Fig. 7A) (Arif et al, 2021). Detection of tension was based on a force sensitive elastic peptide that connected the donor and acceptor fluorophores within the FRET tension module. Force above 4 pN extends the elastic peptide linker resulting in separation of the two fluorophores YPet and mCherry, thereby reducing FRET. To rule

out that force independent effects would influence FRET efficiency of our module, a negative control construct was used. In this construct (VEC-TS-NF, where NF stands for "no force"), the FRET module was fused to the C-terminus of VE-cadherin, which prevented that pulling forces on the catenins would affect FRET efficiency (Fig. 7A).

Using the VE-cadherin tension sensor, we tested whether gene silencing of plakoglobin or β-catenin would affect leukocyte induced tension across VE-cadherin. To this end, we first treated HUVEC with VE-cadherin siRNA plus either control, plakoglobin or β-catenin siRNA for 24 h followed by harvesting and reseeding the cells in flow chambers and transducing them with the VE-cadherin tension sensor (VEC-TS) or the negative control construct VEC-TS-NF. These cells were then either exposed to flow with neutrophils or flow alone. To determine FRET efficiency, donor fluorescence lifetime was measured by fluorescence lifetime imaging microscopy (FLIM). As shown in Fig. 7B, direct comparison of endothelial junctions at sites of PMN transmigration with junctions of HUVEC without PMNs revealed a highly significant 10% drop in FRET efficiency, whereas no such effect was seen when plakoglobin expression was silenced. Neutrophil-induced drop in FRET efficiency was force dependent, since it was not detected with HUVEC expressing the VEC-TS-NF negative control construct (Fig. 7B). We conclude that neutrophil induced increase in tension across VE-cadherin requires the expression of plakoglobin. In contrast to these results, we found that β-catenin siRNA treatment did not interfere with the neutrophil-induced drop in FRET efficiency (Fig. 7C). A representative image of our FLIM measurements is depicted in Fig. 7D, illustrating the increased life-time of donor fluorescence (FLIM) at neutrophil transmigration sites which was detected for control siRNA and β-catenin siRNA treated HUVEC, but not for plakoglobin-siRNA treated cells. Efficiency of plakoglobin and β-catenin silencing is documented in the immunoblot of Fig. 7E.

Collectively, our results show that similar to the selective need of plakoglobin for leukocyte diapedesis, it is only plakoglobin and not β-catenin which is needed for neutrophil induced tension across VE-cadherin. This is in agreement with a concept that tension across VE-cadherin-plakoglobin-α-catenin complexes induced by neutrophils is involved in the neutrophil diapedesis process.

## Vascular permeability inducing factors stimulate tension across VE-cadherin via plakoglobin in vitro

The results described above prompted us to test whether inflammation induced vascular leaks would also trigger tension

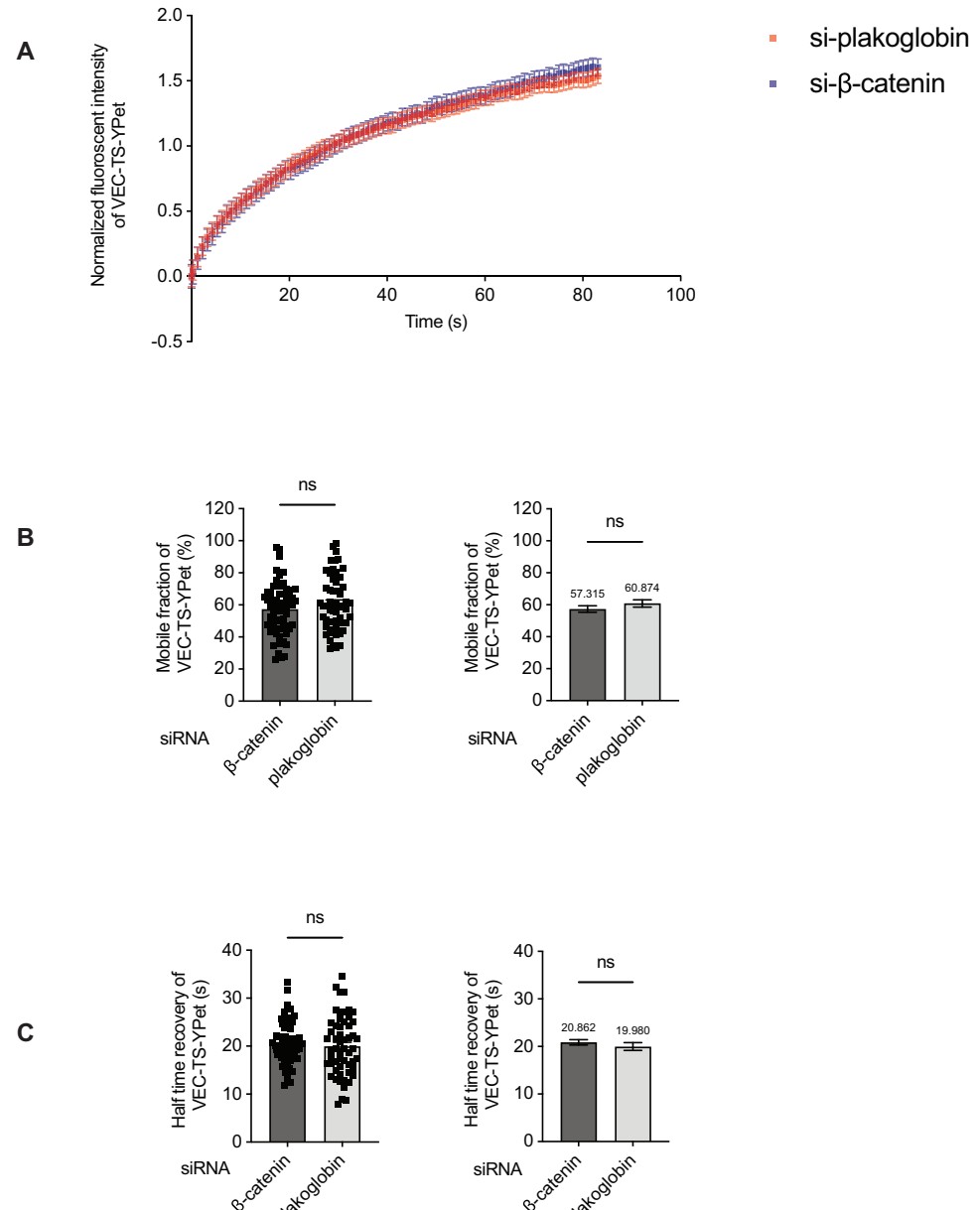

**Figure 6.  Silencing of plakoglobin or β-catenin does not affect VE-cadherin mobility measured by fluorescence recovery after photobleaching (FRAP) at endothelial junctions.**

(**A**) Normalized FRAP recovery curves for HUVEC expressing VEC-TS-YPet adenoviral construct, depleted of endogenous plakoglobin (red, 56 areas analyzed) or β-catenin (blue, 62 areas analyzed) using siRNA. Mean fluorescence intensities within the bleached region of interest (ROI) were background-subtracted, corrected for overall photofading using the reference ROI, and normalized to the pre-bleach intensity (set to 1) and post-bleach intensity (set to 0); a constant offset was subtracted from each curve to correct for small negative baseline values generated by the normalization procedure. (**B, C**) Mobile fraction (**B**) and recovery half-time (**C**) of VEC-TS-YPet in transduced HUVEC treated with siRNA for plakoglobin or β-catenin were calculated from the curve shown in (**A**). In (**B, C**), left graph represents distribution of individual data points whereas right graph shows bar plot of the mean ± SEM with the numerical mean displayed above. Data information: Data from n = 56 (si-plakoglobin) and n = 62 (si-β-catenin) ROI's, pooled from three independent experiments displayed as mean ± SEM. ns, not significant by t-test. Source data are available online for this figure.

across VE-cadherin selectively in a plakoglobin-dependent way. To this end, again we silenced the expression of endogenous VE-cadherin in addition to the treatments with control, plakoglobin or β-catenin siRNA followed by transduction with either the VEC-TS or the negative control VEC-TS-NF construct. These cells were

then exposed for 10 min to histamine or left untreated and FRET efficiency was determined by FLIM measurements of donor fluorescence life time. As shown in Fig. 8A, histamine indeed caused a drop in FRET efficiency, which was also seen in β-catenin siRNA-treated cells, but was completely blocked when plakoglobin

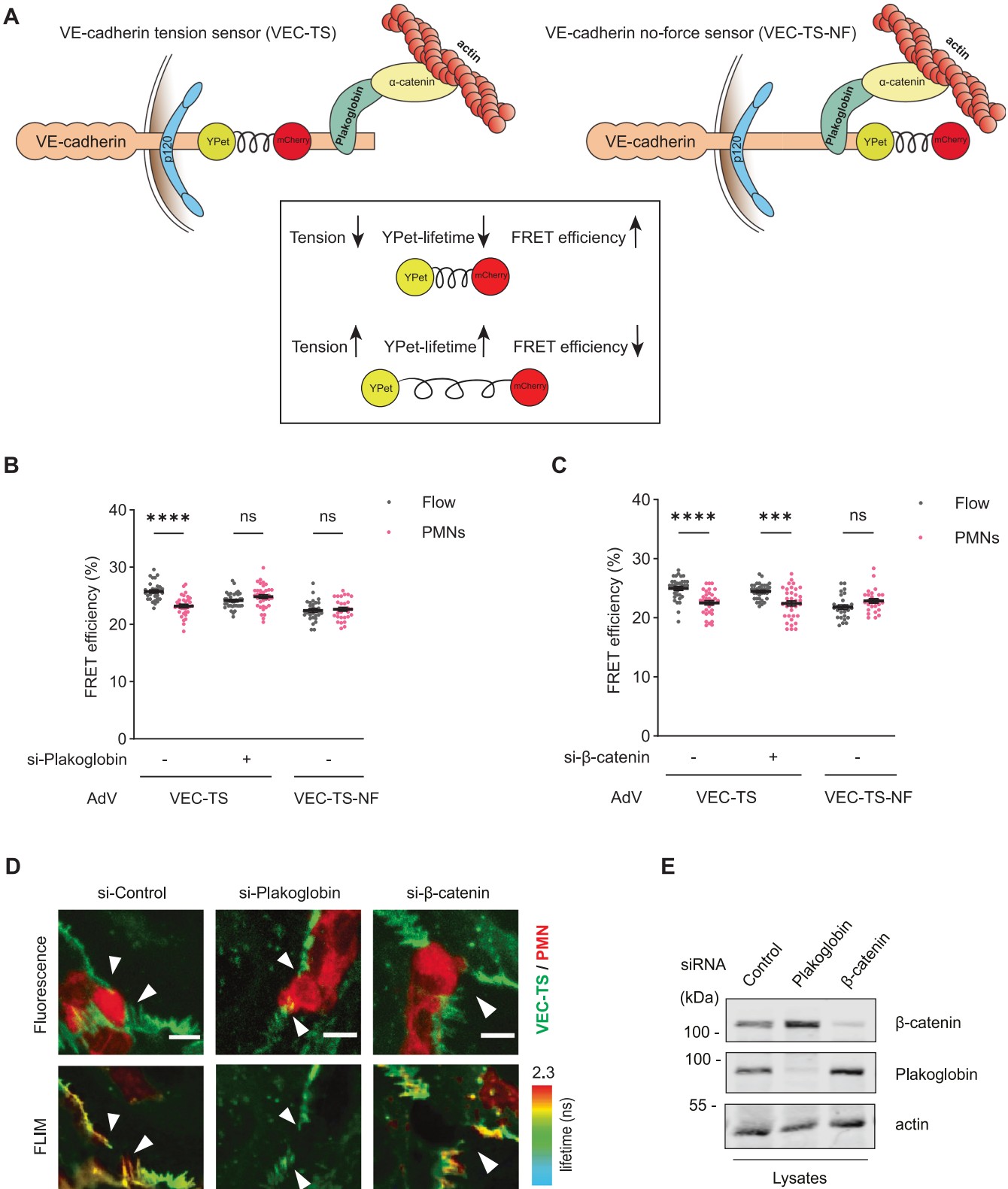

**Figure 7.   Plakoglobin, not β-catenin, is needed for neutrophil induced tension across VE-cadherin.**

(A) Schematic representation of VE-cadherin tension sensor (VEC-TS) and VE-cadherin no-force sensor (VEC-TS-NF). The VE-cadherin tension sensor has a ferredoxin-like (FL) linker-based FRET module inserted within VE-cadherin between the p120 and the β-catenin/plakoglobin binding site, whereas the VE-cadherin no-force sensor has a FRET module inserted before the STOP codon of VE-cadherin. The FRET module (depicted in the box) consists of two fluorophores, YPet and mCherry separated by an elastic linker peptide. When the module is under tension, FRET from YPet to mCherry decreases. (B, C) Quantification of FRET efficiency (%) in junctions of HUVEC expressing VEC-TS or VEC-TS-NF. Cells were pre-treated with control, plakoglobin (B) or β-catenin (C) siRNA, transduced with VEC-TS or VEC-TS-NF adenovirus (AdV) followed by exposure to flow with human PMNs or to flow alone. Cells were fixed with 4% PFA and washed with PBS. FLIM measurements were performed at sites of transmigration (PMNs) or at junctions without PMNs (Flow). (D) Representative images of HUVEC expressing VEC-TS at sites of PMN transmigration (arrowheads), pre-treated with control, plakoglobin or β-catenin siRNA. The upper panel displays a maximum intensity projection of the Z-stack of YPet fluorescence (VEC-TS in green) and CellTracker DeepRed (PMN in red). The lower panel shows the amplitude-averaged lifetime of YPet per pixel (FLIM) of the same cells. Scale bars, 5 µm. (E) Representative immunoblot of lysates from control, plakoglobin or β-catenin siRNA treated HUVEC, detected with antibodies against plakoglobin, β-catenin and actin. Data information: Mean ± SEM of at least 27 measurements per group from three independent experiments (B, C). ***$P = 0.0004$, ****$P < 0.0001$, two-way ANOVA (B, C). ns, not significant. Source data are available online for this figure.

was silenced. No drop in FRET efficiency was detected for the no-force negative control construct (Fig. 8A). A representative illustration of the results is given in Fig. 8C. Since our in vitro effects on endothelial barrier integrity were determined for thrombin (Fig. 3), we next tested whether thrombin-induced tension across VE-cadherin would also require plakoglobin. Comparable to our results with histamine, we found again that the thrombin-induced drop of FRET efficiency was similar for control and for β-catenin siRNA treated cells (Fig. 8B). We could also reproduce the result that plakoglobin siRNA treatment significantly reduced the drop in FRET efficiency, yet with thrombin this drop was not completely blocked (Fig. 8B). Thus, in contrast to histamine treated cells, the lack of plakoglobin could not completely block the induction of some tension across VE-cadherin by thrombin. Despite this difference, our results clearly show that plakoglobin is of vital importance for the generation of tension across VE-cadherin-catenin complexes at endothelial junctions by inflammatory stimuli, whereas β-catenin is dispensable.

### Variability of FRET efficiency of the VE-cadherin no force construct in HUVEC grown in different chamber slides

When comparing the results of our FLIM/FRET measurements shown in Figs. 7 and 8, it became apparent that baseline FRET efficiency of the NF-construct varied when determined in fibronectin-coated flow chambers and in fibronectin-coated 8-well chambers. Since the experiments in flow chambers were performed with cells pre-treated with TNF-α and cells had been incubated with Ca$^{2+}$/Mg$^{2+}$-buffer (containing BSA) under flow prior to fixation, we directly compared all of these culture conditions when determining FRET efficiency of the VEC-TS-NF construct in HUVEC. As shown in Fig. EV4, we found that FRET efficiency indeed varied due to having the cells in the two different types of chambers, whereas FRET efficiency was independent of TNF-α stimulation or of the type of buffer or medium. Interestingly, we made similar observations for an alternative VE-cadherin-NF construct, where the tension sensor was at the same position in VE-cadherin as in the force construct, while the 3' C-terminus of VE-cadherin was deleted in order to prevent catenin interactions (Arif et al, 2021). This NF-construct was designed in the same way as the original NF-construct by the Schwartz lab (Conway et al, 2013). Although the two NF-constructs carry the sensor in a different position within the cytoplasmic domain of VE-cadherin, they have in common that the sensor is located at the

very C-terminus of a fusion protein. We do not know why these constructs show a lower baseline FRET efficiency in the flow chambers than the force sensitive construct. Independent of this phenomenon, however, our results show that both VE-cadherin-NF constructs allow valid no-force control measurements, since they are not sensitive to signaling induced by thrombin, histamine or neutrophils.

### Generation of knock-in mice expressing a VE-cadherin tension sensor replacing endogenous VE-cadherin

This far, the selective role of plakoglobin in processes that interfere with junction integrity in vitro and in vivo was in full agreement with its selective role for the induction of tension across VE-cadherin in cultured endothelial cells. In order to test, first, whether a rise in tension across VE-cadherin could also be observed in vivo under inflammatory conditions and, second, whether plakoglobin is again involved in such effects, we decided to generate knock-in mice expressing a VE-cadherin tension sensor. To this end we used a recombinase-mediated cassette exchange (RMCE) approach (Fig. 9A) which we used previously to replace endogenous VE-cadherin by VE-cadherin mutants (Schulte et al, 2011; Broermann et al, 2011; Wessel et al, 2014; Wilkens et al, 2024; Holtermann et al, 2025). We generated separate mouse lines for each of three constructs, the VE-cadherin tension sensor (VEC-TS) and the negative control "no force" construct which was not sensitive to catenin-driven force on VE-cadherin (VEC-TS-NF), as they are described in Fig. 7A. In addition, we generated a third knock-in mouse with a "donor-only" VE-cadherin construct that looked identical to the VEC-TS construct but carried a mutated acceptor fluorophore. This construct was used to measure the donor life time in the absence of any possible FRET influence. Expression levels of the VEC-TS, the VEC-TS-NF and the VEC-TS-YPet fusion proteins at endothelial cell contacts were identical to endogenous VE-cadherin in C57Bl6 mice and to VE-cadherin in knock-in mice carrying a cDNA of WT-VE-cadherin in the VE-cadherin locus, as was determined by whole-mount staining of cremaster muscle with anti-VE-cadherin antibodies (Fig. 9B).

### Histamine induces tension across VE-cadherin in cremaster venules in vivo and plakoglobin is needed

Making use of our VE-cadherin tension sensor knock-in mice, we next tested whether histamine can induce tension across VE-

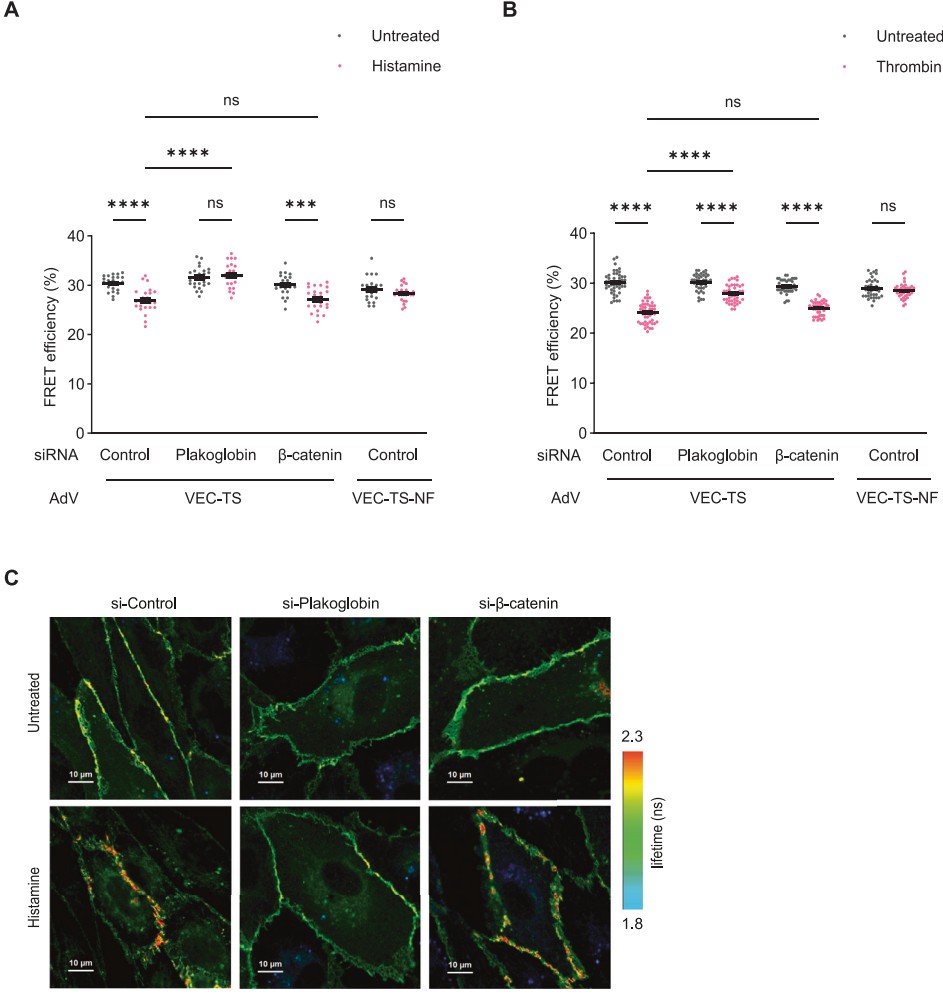

**Figure 8. Inflammatory mediators stimulate tension across VE-cadherin via plakoglobin in vitro.**

(**A, B**) Quantification of FRET efficiency (%) in junctions of HUVEC expressing VEC-TS or VEC-TS-NF, transfected with control, plakoglobin or β-catenin siRNA, after 10 min of histamine (100 μM) (**A**), or thrombin (1 U/mL) (**B**) stimulation and in untreated conditions. (**C**) Representative FLIM (amplitude-averaged lifetime of YPet per pixel) images of HUVEC expressing VEC-TS, transfected with control, plakoglobin or β-catenin siRNA, after histamine stimulation or in untreated conditions. Scale bars, 10 μm. Data information: Mean ± SEM of 24 measurements per group (**A**), or at least 38 measurements per group (**B**) pooled from three independent experiments. ***P = 0.0001, ****P < 0.0001, two-way ANOVA (**A, B**). ns, not significant. Source data are available online for this figure.

cadherin in vivo. While it is often assumed that inflammatory mediators may cause pulling forces on endothelial junctions, this was never shown in vivo. Furthermore, it was questioned whether actin stress fibers which are potentially able to mediate pulling and separation of endothelial junctions in vitro would exist in leaky vessels in vivo (Claesson-Welsh et al, 2021). To test, whether histamine would increase tension across VE-cadherin in venules where it stimulates leakage, we co-injected histamine or PBS together with 20 nm fluorescent microspheres i.v. for 3 min into VEC-TS and VEC-TS-NF knock-in mice before collecting and fixing parts of the cremaster muscle for FLIM analysis. As illustrated in Fig. 10A histamine caused leaks in venules in mice of both genotypes as was detected by accumulation of fluorescent microspheres at endothelial junctions which was not seen when PBS was co-injected with the microspheres. Quantification of fluorescent microsphere accumulation revealed no significant

difference between mice expressing the VEC-TS construct and those expressing the VEC-TS-NF control (Fig. 10B). When the same venules were analyzed by FLIM measurements, we found that histamine treatment increased the lifetime of YPet signals (which means: decreased FRET efficiency) in the VEC-TS knock-in mouse but not in the VEC-TS-NF negative control mouse (Fig. 10C). No effect was seen in venules of PBS-injected mice. Quantitation of FLIM analysis was performed for two mice (over 16 venules) of the F1 generation (Fig. 10D) and for 3 mice (over 10–15 venules) of the F6 generation (Fig. 10E) of transgenic knock-in mice. In both series of experiments, histamine clearly and significantly reduced FRET efficiency by 9.5% (F1 mice) and 19.8% (F6 mice) in VEC-TS mice whereas no drop was seen in VEC-TS-NF mice. We conclude, that histamine induces mechanic tension across VE-cadherin at endothelial junctions of leaky venules in the cremaster muscle.

**A**

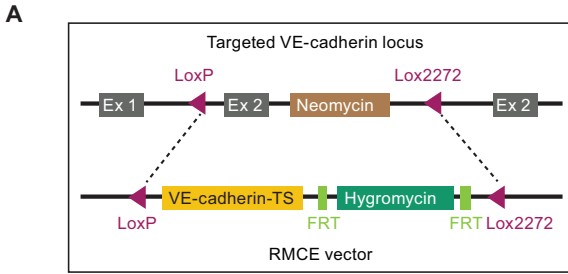

**B**

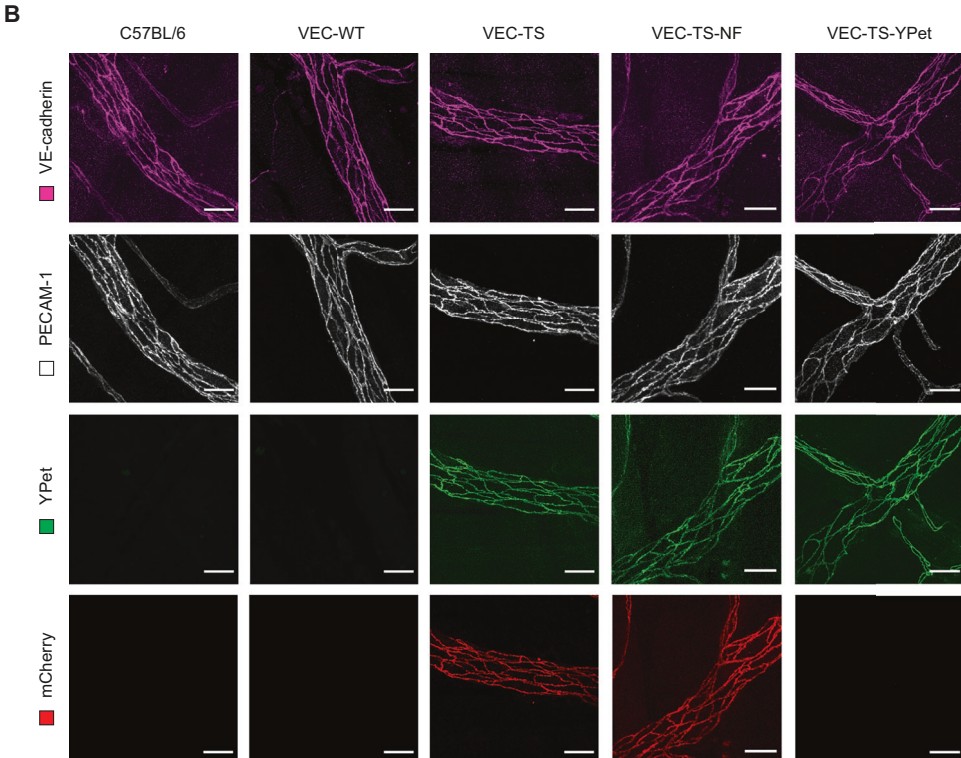

**Figure 9. Characterization of knock-in mice expressing VE-cadherin tension sensor.**

(**A**) Recombinase-mediated cassette exchange (RMCE) used to target exon 2 (Ex2) of the VE-cadherin gene locus, which contains the ATG start codon. The strategy involves flanking exon 2 with two incompatible loxP sites (loxP and lox2272). Cre recombinase facilitates the replacement of exon 2 with a loxP/lox2272-flanked cassette containing cDNA for the VE-cadherin tension sensor followed by a polyA-transcriptional stop cassette and a hygromycin gene (for selection) flanked by FRT sites. (**B**) Representative images of whole-mounts of cremaster muscle from wild-type C57BL/6 mice and homozygous VEC-WT, VEC-TS, VEC-TS-NF and VEC-TS-YPet knock-in mice, with VE-cadherin and PECAM-1 immunostaining in upper panels and endogenous YPet and mCherry signals in lower panels, presented as maximum intensity projections of Z-stacks. Scale bars, 25 μm. Source data are available online for this figure.

Next, we asked whether plakoglobin is required in vivo for histamine induced tension across VE-cadherin. To this end, we bred our VEC-TS mice with *Jup*$^{lox/lox}$ and with *Jup*$^{ECKO}$ mice. As shown in immunoblots, gene inactivation of plakoglobin in VEC-TS- *Jup*$^{ECKO}$ mice was highly efficient (Fig. 11A). Quantitation of FLIM analysis over at least 16 vessels from 3 mice per group revealed that histamine-induced drop of FRET efficiency was only observed in mice expressing endothelial plakoglobin, but was not detected in mice that had lost plakoglobin (Fig. 11B). Collectively, we could reproduce our in vitro results about the requirement of plakoglobin for the induction of tension across VE-cadherin by histamine in vivo.

## Discussion

Plakoglobin and β-catenin each link VE-cadherin in two alternative, co-existing complexes to the actin cytoskeleton. The purpose of this study was to find out whether each of these catenins is of similar importance for the regulation of endothelial junctions in inflammation and for the way actin linkage affects VE-cadherin in this process. Based on gene silencing in HUVEC and endothelial selective gene inactivation in mice, we found that only plakoglobin, but not β-catenin was required for inflammation-induced neutrophil diapedesis and vascular leak formation. Investigating whether blocking the expression of plakoglobin or

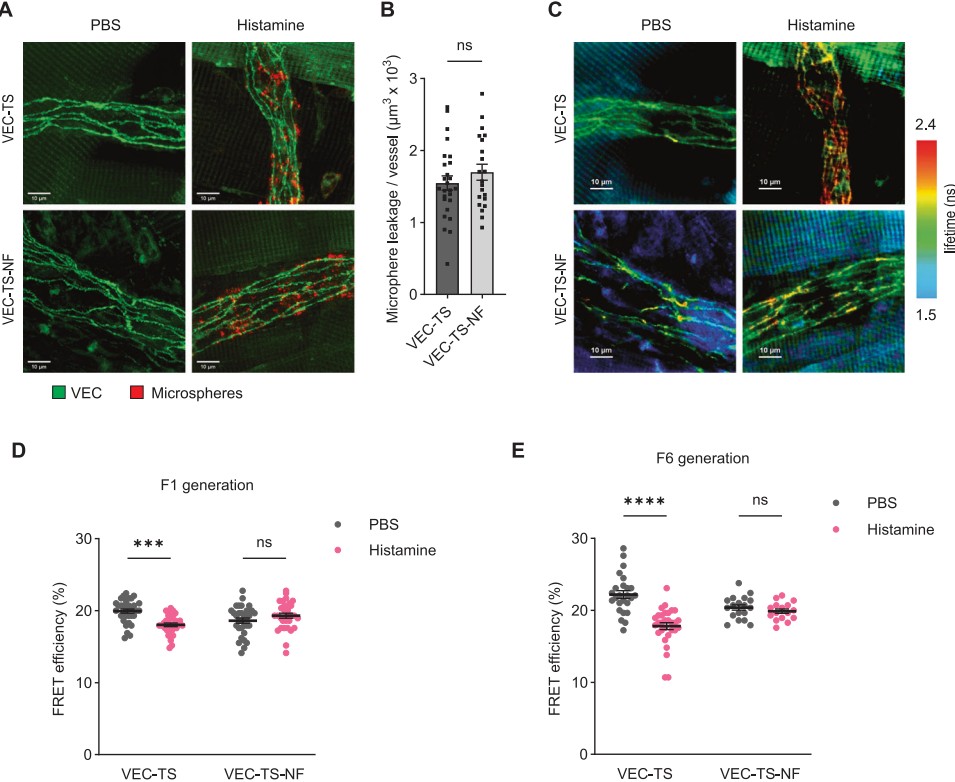

**Figure 10. Vascular permeability induction increases tension across VE-cadherin in cremaster venules.**

(A–E) Whole-mounts of cremaster muscle from VEC-TS or VEC-TS-NF mice given intravenous injection of PBS or histamine together with 20 nm fluorescent microspheres for 3 min before collection and fixation of tissue. (A) displays a representative maximum intensity projection of a Z-stack of YPet fluorescence (VE-cadherin in green) and crimson FluoSpheres (fluorescent microspheres in red), (B) shows quantification of microspheres in histamine-stimulated VEC-TS and VEC-TS-NF mice and (C) shows the amplitude-averaged lifetime of YPet per pixel of the same venules as in (A). Scale bars, 10 μm. Graph (D, E) represents quantification of FRET efficiency (%) in endothelial junctions of cremaster venules from mice of F1 generation (D) and F6 generation (E). Data information: Mean ± SEM from 4 mice per group, with a total of 26 (VEC-TS) and 21 (VEC-TS-NF) vessels analyzed (B) or mean ± SEM of 32 measurements per group pooled from two independent experiments (D) or (n = 25, 30, 18, 16) measurements pooled from three independent experiments (E) (one mouse per experiment in D, E). Unpaired t-test (B), ***P = 0.0002, ****P < 0.0001, two-way ANOVA (D, E). ns, not significant. Source data are available online for this figure.

β-catenin would affect VE-cadherin, we found, that silencing of each of the catenins was fully compensated by upregulation of the other catenin in VE-cadherin complexes. Despite of this, only the lack of plakoglobin, but not of β-catenin, blocked leukocyte-induced and inflammatory mediator-induced tension across VE-cadherin. The generation of knock-in mice by replacing endogenous VE-cadherin with a FRET-based VE-cadherin tension sensor allowed us to verify these results for histamine in vivo. Indeed, we could show that histamine does trigger tension across VE-cadherin at endothelial junctions in inflamed venules and this effect is dependent on the presence of plakoglobin. Thus, plakoglobin provides the necessary linkage of VE-cadherin to actin that is needed to induce tension across VE-cadherin during inflammation, and β-catenin cannot replace this function.

The catenins were identified as important cytosolic molecules that link the cytoplasmic tail of cadherins to the circumferential actin cytoskeleton thereby providing anchorage and mechanical support to cadherins which rather maintain than challenge adhesive interactions in trans with cadherin molecules on the neighboring cell (Ozawa et al, 1989). This first study showed that

catenins strongly support the adhesive interaction of cadherins. On the other hand, catenins are needed to exert actomyosin induced force across VE-cadherin (Conway et al, 2013), a phenomenon which is linked to the transmigration of neutrophils through endothelium (Arif et al, 2021). Here, we show that also inflammatory mediators such as histamine or thrombin, well-known to induce vascular leaks, induce tension across VE-cadherin, and we showed this here not only in vitro but even in vivo. This is not trivial, since it has been debated whether gap formation between endothelial cells in the context of leak formation requires cytoskeletal stress fibers that increase tension at junctions (Claesson-Welsh et al, 2021), simply because such stress fibers, which are well documented in cultured endothelial cells, could not be visualized in endothelium of leaky vessels (Adamson et al, 2003; Waschke et al, 2004). This raises the question which type of actin structures are mediating the pulling on VE-cadherin. It is possible that actomyosin fibers which exert tension across VE-cadherin have more subtle structures than the huge stress fibers that are usually detected in cultured endothelial cells connecting focal adhesions. In addition, we speculate that focal contacts in the vicinity of junctions

**A**

**B**

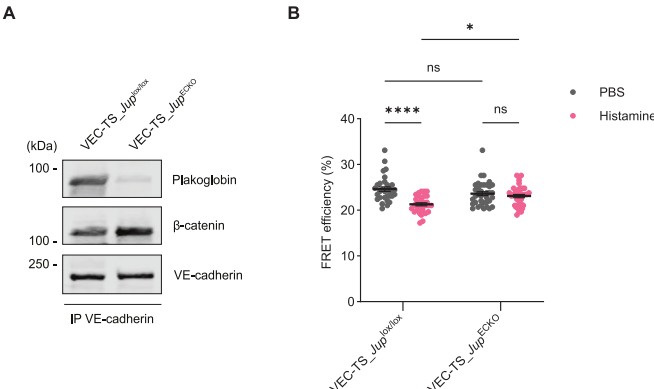

**Figure 11. Plakoglobin is indispensable for histamine induced tension across VE-cadherin in cremaster venules.**

(A) Representative immunoblot of VE-cadherin immunoprecipitates from lung lysates of VEC-TS_ *Jup*lox/lox or VEC-TS_*Jup*ECKO mice, detected with antibodies against plakoglobin, β-catenin and VE-cadherin. (B) Quantification of FRET efficiency (%) in endothelial junctions of cremaster venules from VEC-TS_ *Jup*lox/lox or VEC-TS_*Jup*ECKO mice given intravenous injection of PBS or histamine together with 20 nm fluorescent microspheres for 3 min before collection and fixation of tissue. Data information: Mean ± SEM of at least 32 measurements per group pooled from three independent experiments (one mouse per experiment) (B). *$P = 0.0135$, ****$P < 0.0001$, two-way ANOVA. ns, not significant. Source data are available online for this figure.

could be anchor points for actomyosin structures pulling on junctions. Close vicinity to focal contacts would probably make it difficult to detect such structures in vivo.

The induction of tension across VE-cadherin raises the question how this tension would contribute to increasing junction permeability and leukocyte diapedesis. Since both processes are supported by force exerted on VE-cadherin, we assume that the contribution is of some general nature. We speculate that the increase of mechanical force will challenge the trans-interaction between VE-cadherin molecules from adjacent cells. This will lead on average to a higher percentage of cadherin molecules which are disengaged and therefore available for endocytosis. Since vascular permeability induction and leukocyte diapedesis both influence tyrosine phosphorylation and ubiquitination of VE-cadherin which supports the endocytosis process of VE-cadherin (Wessel et al, 2014; Wilkens et al, 2024; Holtermann et al, 2025; Arif et al, 2021), synergistic effects can be expected if VE-cadherin interactions are exposed to tension and to endocytosis driving mechanisms.

It is intriguing, that plakoglobin is on the one hand able to support VE-cadherin function (Lampugnani et al, 1995; Schnittler et al, 1997; Muramatsu et al, 2017; Hamad et al, 2025) and on the other hand it is able to challenge it by mediating the link to tension-increasing actomyosin structures. Since on the protein expression level there was full compensation for the loss of one catenin by the other (in vitro and in vivo), the dramatic effects we observed for the loss of plakoglobin clearly document that β-catenin cannot efficiently fulfill the functions of plakoglobin, which are needed to exert force across VE-cadherin in the context of inflammation and to support junction permeability and leukocyte diapedesis. This specific role of plakoglobin in challenging junctions is in agreement with a previous study where we reported that plakoglobin is a preferred substrate among the catenins for the

tyrosine phosphatase VE-PTP. This phosphatase dissociates from the VE-cadherin-catenin complex during leukocyte extravasation and vascular permeability induction, which leads to the phosphorylation of VE-cadherin and preferentially of plakoglobin (Nottebaum et al, 2008). This dissociation of VE-PTP from VE-cadherin is even required for leukocyte extravasation and vascular permeability induction in vivo (Broermann et al, 2011). Since VE-PTP dissociation inhibited the adhesive function of VE-cadherin and this negative effect was dependent in part on plakoglobin, we speculated that plakoglobin could be involved in mechanisms challenging junctions under inflammatory conditions (Nottebaum et al, 2008).

Other possibilities to explain the specific function of plakoglobin in challenging junctions could be that plakoglobin may help to recruit factors needed for the activation of myosin so that only filaments connected via plakoglobin to VE-cadherin stimulate tension across VE-cadherin. It is well documented that myosin activation at radial stress fibers and circumferential actin are independently regulated (Ando et al, 2013). In addition, it was shown that JAM-A and ZO-1 represent determinants which decide on the recruitment of active myosin II to either circumferential actin or stress fibers which affect tension across VE-cadherin (Tornavaca et al, 2015). It was shown that fluid leakage during leukocyte transmigration is prevented by the formation of contractile F-actin structures that surround transmigrating neutrophils and keep the emigration pore tight (Heemskerk et al, 2016). In the light of our results shown here, this suggests, that force is generated on both types of actomyosin structures, those that support the opening of junctions via affecting VE-cadherin as well as others that tighten the pore around the transmigrating neutrophil. Each of these processes may be driven by different signaling mechanisms, in line with previous studies (Ando et al, 2013).

Whether plakoglobin indeed contributes to fine tuning the spatial organization of actomyosin activation at junctions will need to be analyzed in the future. Likewise, high-resolution microscopy to image the distribution of plakoglobin and β-catenin at junctions may be interesting. So far, with regular confocal microscopy no differences are detectable.

Only very few studies have investigated differences between the function of β-catenin and plakoglobin at endothelial junctions. Junctional recruitment differs during maturation of junctions, with plakoglobin being recruited later than β-catenin (Lampugnani et al, 1995; Schnittler et al, 1997). Whether histamine, thrombin or neutrophils differentially affect the presence or distribution of the two catenins at junctions is unknown. Using high-resolution confocal microscopy, we did not find obvious differences (Fig. EV3). Desmoplakin and the desmosomal cadherin desmoglein were described as specific binding partners for plakoglobin, yet whether desmoplakin does indeed bind to VE-cadherin associated plakoglobin was only indirectly shown (Kowalczyk et al, 1998). We are not aware of any evidence that suggests a role for desmoglein-2 in endothelial junction control during inflammation. The preferential effects of VE-PTP on plakoglobin might be a hint towards selective ways to address the function of this catenin as regulator of endothelial junctions (Nottebaum et al, 2008). Whether tyrosine phosphorylation of plakoglobin or β-catenin affects binding to VE-cadherin is controversial, with some evidence reported for β-catenin dissociation (Chen et al, 2012; Monaghan-Benson and

Burridge, 2009), whereas plakoglobin phosphorylation was suggested to destabilize the binding to α-catenin (Miravet et al, 2003). Others, however, reported that cadherin-catenin complexes remain intact during destabilization of junctions by various mediators (Andriopoulou et al, 1999; Esser et al, 1998; Nottebaum et al, 2008; Timmerman et al, 2012). Future studies will be needed to elucidate whether differential modifications or specific binding partners enable the differential roles of plakoglobin and β-catenin for the control of endothelial junctions and VE-cadherin function.

Our in vivo studies were facilitated by the fact that endothelial-specific gene inactivation of plakoglobin resulted in viable *Jup^ECKO* mice with no obvious defects. In contrast, *Ctnnb1^iECKO* are not viable and show seizures within two weeks after tamoxifen treatment due to blood–brain barrier breakdown caused by the loss of claudins (Tran et al, 2016). Therefore, we had to limit the duration of tamoxifen treatment and could not completely deplete the expression of β-catenin. While this limits the value of the results with these mice, it is important to note that gene inactivation reduced the expression of plakoglobin by 93% in *Jup^ECKO* mice which was quantitatively compensated by upregulation of β-catenin by 84%, thus demonstrating that β-catenin cannot functionally replace plakoglobin, since otherwise *Jup^ECKO* would not have shown the deficits which we found. Therefore, we believe our results with *Jup^ECKO* mice clearly show that only plakoglobin supports vascular permeability induction, neutrophil diapedesis and tension across VE-cadherin triggered in these processes, whereas β-catenin cannot provide these functions. This conclusion was also fully supported by our in vitro results.

The first FRET-based VE-cadherin tension sensor was designed and used for measuring effects of flow-induced shear stress by the Schwartz group (Conway et al, 2013). This construct was also expressed in zebrafish to analyze changes in junctional tension during embryonic development of the vasculature (Lagendijk et al, 2017). We used a similarly designed construct in a previous study on leukocyte diapedesis (Arif et al, 2021), with the only difference that the force sensitive peptide linking the two fluorophores was changed to a ferredoxin-like peptide (Austen et al, 2015; Ringer et al, 2017). The VEC-TS construct we used in the present study was identical to the one we used before. Only our negative control, no-force construct (VEC-TS-NF) was designed in a different way. This construct is needed as a negative control to verify that any change in FRET efficiency that is detected with the VEC-TS is not seen with the VEC-TS-NF construct and thus is indeed due to force exerted on VE-cadherin via the catenin complex and not due to other changes, for example in the chemical environment. Previously, the no-force construct contained a truncated form of VE-cadherin lacking the β-catenin-binding region. Since mice with such a deletion in VE-cadherin are not viable, we changed this design in order to enable the generation of knock-in mice. Instead of inserting the tension-module between the p120 and the β-catenin-binding site, we simply fused the module to the C-terminus of VE-cadherin, thereby avoiding that force across the catenins could pull on the tension sensor. This no-force construct was used in the present study for our in vitro and the in vivo studies.

We would have liked to make use of our VEC-TS knock-in mice to investigate in vivo the effects of neutrophils on triggering tension across VE-cadherin. Analysis of donor fluorescence lifetime using FLIM of endothelium in cremaster venules of living mice at sites where neutrophils were extravasating could not be done, since a simultaneous visualization of the transmigration process by LSM and FLIM analysis was not possible. Therefore, we analyzed longer stretches of the endothelium of venules which interacted with numerous neutrophils. The tissue was fixed with paraformaldehyde to allow longer observation time (sufficient photon counts). However, in such fixed samples it turned out to be too difficult to identify a sufficient number of neutrophils for which we could unambiguously determine whether they were in the process of transmigration or simply attaching to the endothelial cell surface. Therefore, we were not able to measure in vivo whether neutrophil transmigration would increase tension across VE-cadherin. However, it was shown before by staining cremaster tissue with antibodies against phosphorylated myosin light chain that actomyosin-generated tension is stimulated during neutrophil diapedesis (Heemskerk et al, 2016).

In summary, our results reveal an important difference between plakoglobin and β-catenin in inflammation with plakoglobin being exclusively relevant as supporter of leukocyte diapedesis, the induction of vascular permeability and the stimulation of mechanical tension across VE-cadherin in this context.

# Methods

**Reagents and tools table**

| Reagent/Resource | Reference or Source | Identifier or Catalog Number |
|---|---|---|
| **Experimental models** | | |
| Human umbilical vein endothelial cells (HUVEC) (*H. sapiens*) | In-house preparation | N/A |
| Polymorphonuclear leukocytes (PMNs) (*H. sapiens*) | In-house preparation | N/A |
| 29A4 embryonic stem (ES) cells (*M. musculus*) | (Hooper et al 1987) | N/A |
| HEK-293A cells (*H. sapiens*) | Invitrogen | K4930-00 |
| C57BL/6 (*M. musculus*) | Janvier Labs | N/A |
| SV129 (*M. musculus*) | Janvier Labs | N/A |

| Reagent/Resource | Reference or Source | Identifier or Catalog Number |
|---|---|---|
| **Recombinant DNA** | | |
| pENTR2B | Invitrogen | A10465 |
| pENTR2B β-catenin (*H. sapiens*) | This study | N/A |
| pENTR2B VEC-TS (*H. sapiens*) | (Arif et al, 2021) | N/A |
| pENTR2B VEC-TS-NF (*H. sapiens*) | This study | N/A |
| pENTR2B VEC-TS-YPet (*H. sapiens*) | This study | N/A |
| pAd/CMV/V5-DEST | Invitrogen | V49320 |
| pAd/CMV/V5-DEST β-catenin (*H. sapiens*) | This study | N/A |
| pAd/CMV/V5-DEST VEC-TS (*H. sapiens*) | (Arif et al, 2021) | N/A |
| pAd/CMV/V5-DEST VEC-TS-NF (*H. sapiens*) | This study | N/A |
| pAd/CMV/V5-DEST VEC-TS-YPet (*H. sapiens*) | This study | N/A |
| pAd/CMV/V5-DEST LacZ | This study | |
| RMCE vector VEC-TS (*M. musculus*) | This study | N/A |
| RMCE vector VEC-TS-NF (*M. musculus*) | This study | N/A |
| RMCE vector VEC-TS-YPet (*M. musculus*) | This study | N/A |
| **Antibodies** | | |
| Mouse anti-β-catenin | BD Biosciences | 610154 |
| Mouse anti-γ-catenin | BD Biosciences | 610254 |
| Mouse anti-actin | BD Biosciences | 612656 |
| Mouse anti-alpha-tubulin | Sigma-Aldrich | T6074 |
| Goat anti-VE-cadherin | R&D Systems | AF1002 |
| Goat anti-VE-cadherin | R&D Systems | AF938 |
| Rabbit anti-VE-cadherin | In-house preparation (Gotsch et al, 1997) | C5 |
| Rabbit anti-VE-cadherin | In-house preparation (Broermann et al, 2011) | VE42 |
| Rabbit anti-γ-catenin | In-house preparation This study | |
| Rat anti-PECAM1 | In-house preparation (Wegmann et al, 2006) | 1G5.1 |
| Donkey anti-mouse IRDye 680RD | LI-COR Biosciences | 926-68072 |
| Donkey anti-mouse IRDye 800CW | LI-COR Biosciences | 926-32212 |
| Donkey anti-mouse HRPO | Jackson ImmunoResearch | 715-035-150 |
| Donkey anti-mouse AlexaFluor 568 | Invitrogen | A10037 |
| Donkey anti-goat IRDye 680RD | LI-COR Biosciences | 926-68074 |
| Donkey anti-goat IRDye 800CW | LI-COR Biosciences | 926-32214 |
| Donkey anti-goat HRPO | Jackson ImmunoResearch | 705-035-147 |
| Donkey anti-goat AlexaFluor 488 | Invitrogen | A11055 |
| Donkey anti-rabbit AlexaFluor 488 | Invitrogen | A21206 |
| Donkey anti-rabbit AlexaFluor 405 | Invitrogen | A48258 |
| Donkey anti-rabbit AlexaFluor 647 | Invitrogen | A31573 |
| Donkey anti-rat AlexaFluor 647 | Invitrogen | A78947 |

| Reagent/Resource | Reference or Source | Identifier or Catalog Number |
|---|---|---|
| **Oligonucleotides and other sequence-based reagents** | | |
| β-catenin siRNA (5′-AGAAUUGAGUAAUGGUGUAUU-3′) | Qiagen | |
| Plakoglobin siRNA (5′-CCAUCGGCUUGAUCAGGAAUU-3′) | Qiagen | |
| VE-cadherin siRNA (5′-GGGUUUUUGCAUAAUAAGCUU-3′) | Ambion | |
| Mutagenic primer-Gly668 (5′-GCTTAATTCTGTGCGCGGTGTCGACGTCTATACGCGTGGCTCCACTAAGCCCCTGC-3′) | This study | |
| Mutagenic primer-STOP-codon (5′-CCAGGAGGAACTCATCATCGTCGACGTCTATACGCGTTAGGGTTCTGGTCTTTGGG-3′) | This study | |
| Mutagenic primer-Tyr72 (5′-GTAGGCCTTGGAGCCGAGCATGAACTGAGGGGAC-3′) | This study | |
| PCR forward primer (5′-GGGTCGACATGGTGAGCAAAGGCGAAGAGC-3′) | This study | |
| PCR reverse primer (5′-TTACGCGTCTTGTACAGCTCGTCCATGCC-3′) | This study | |
| **Chemicals, Enzymes and other reagents** | | |
| Dispase II | Roche | 4942078001 |
| EBM-2 medium | Lonza | CC-3156 |
| Histopaque®-1077 | Sigma-Aldrich | 10771 |
| Histopaque®-1119 | Sigma-Aldrich | 11191 |
| INTERFERin | Polyplus | 101000016 |
| Lipofectamine RNAiMAX Transfection reagent | Invitrogen | 13778150 |
| Tamoxifen | Sigma-Aldrich | T5648-5G |
| SalI | New England Biolabs | R3138S |
| MluI | New England Biolabs | R3198L |
| Il-1β | BIOMOL GmbH | 50441 |
| Evans blue solution | Sigma-Aldrich | E2129 |
| Histamine | Sigma-Aldrich | H7250-10MG |
| Formamide | Sigma-Aldrich | 11814320001 |
| TNF-α | Sigma-Aldrich | SRP3177-50UG |
| CellTracker Deep Red | Thermo Scientific | C34565 |
| PBS (containing $Ca^{2+}/Mg^{2+}$) | PAN Biotech | P04-35500 |
| Penicillin/streptomycin | PAN Biotech | P06-07100 |
| Fetal bovine serum | Gibco | 10270-106 |
| Fibronectin | Sigma-Aldrich | F1141 |
| cOmplete™, Protease Inhibitor Cocktail Tablets | Roche | 11697498001 |
| Protein A Sepharose CL-4B | Cytiva | 17-0780-01 |
| Paraformaldehyde | Sigma-Aldrich | P6148 |
| Ovalbumin | Sigma-Aldrich | A5503 |
| Bovine serum albumin | Capricorn-Scientific | BSA-1T |
| Thrombin | Calbiochem | 605195-1000U |
| Triton X-100 | Sigma-Aldrich | X-100 |
| Dako Fluorescence mounting medium | Agilent Technologies | S3023 |
| HEPES | Sigma-Aldrich | H0887 |
| OptiMEM I (1X) + GlutaMAX-I | Gibco | 51985-026 |
| FluoSpheres (Fluorescent microspheres) | Invitrogen | F8782 |

| Reagent/Resource | Reference or Source | Identifier or Catalog Number |
|---|---|---|
| **Software** | | |
| Zen 2.3 SP1 Black (64-bit) software | Zeiss | |
| Zen 2 Black edition version 10.0.0.910, licensed ZEN Desk, basic | Zeiss | |
| Graphpad Prism 10 | GraphPad Software Inc. | |
| SymPhoTime 64 (v2.4) | PicoQuant | |
| Fiji-ImageJ | (Schindelin et al, 2012) | |
| Image Studio | LI-COR Biosciences | |
| **Other (Kits, Instruments, lab ware, etc.)** | | |
| Gateway LR Clonase II Enzyme mix | Invitrogen | 11791020 |
| NucleoSpin Gel and PCR Clean-up Kit | Macherey-Nagel | 740609.50 |
| NucleoSpin Plasmid Easy Pure | Macherey-Nagel | 740727.50 |
| QuickChange Lightning Site-directed Mutagenesis Kit | Agilent Technologies | 210519 |
| Rapid DNA Ligation Kit | Thermo Scientific | K1422 |
| ViraPower™ Adenoviral Expression System | Invitrogen | K4930-00 |
| UV-1900i spectrophotometer | Shimadzu | |
| LSM880 | Zeiss | |
| LSM980 | Zeiss | |
| CellBIND | Corning | 430293 |
| Ibidi VI (0.4) flow slides | ibidi | 80601 |
| 8-well µ-Slides | ibidi | 80827 |
| Electrode arrays | Applied BioPhysics | 8W10E |
| ECIS Model 9600 Controller | Applied BioPhysics | |
| Precellys Evolution homogenizer | Precellys | 02520-300-RD000 |
| CKMix 2 ml tubes | Precellys | P000918-LYSK0-A |
| Nitrocellulose membranes | Cytiva | GE10600002 |
| Curix 60 film developer | Agfa | |
| Odyssey Fc imaging system | LI-COR Biosciences | |
| Odyssey CLx imaging system | LI-COR Biosciences | |

## Cell culture

Human umbilical vein endothelial cells (HUVEC) were isolated from umbilical cords (Ethics Committee of Münster University Clinic Approval 2009–537-f-S) by treatment with 1 unit/ml Dispase II (Roche) for 10 min at 37 °C in M199 medium containing 1% penicillin/streptomycin, 20% fetal bovine serum (FBS), 100 µg/ml heparin, and 3.1 µg/ml fungizone. HUVEC were maintained in EBM-2 medium supplemented with EGM-2 MV SingleQuots (Lonza) and cultured on Corning® CellBIND® dishes at 37 °C and 5% $CO_2$. Human PMNs were isolated from blood derived from healthy donors (with formal consent) via density gradient centrifugation (Histopaque®-1077, Histopaque®-1119).

## RNA-mediated interference

Expression of plakoglobin or β-catenin in HUVEC was silenced by transfection with JUP siRNA (5′-CCAUCGGCUUGAUCAGGAATT-3′, Qiagen) or CTNNB1 siRNA (5′-AGAAUUGAGUAAUGGU-GUATT-3′, Qiagen), respectively. Expression of endogenous VE-cadherin in HUVEC was silenced by transfection with CDH5 siRNA (5′-GGGUUUUUGCAUAAUAAGCTT-3′, Ambion). As a negative control, AllStars Negative Control siRNA (Qiagen) was used, which does not share sequence homology with any known mammalian gene. HUVEC were transfected at 60–80% confluency with 20–60 nM siRNA for 72 h using INTERFERin (Polyplus) or Lipofectamine RNAiMAX Transfection reagent (Invitrogen) according to manufacturer's instructions.

## Antibodies

The following commercial and previously generated antibodies were used (IF, immunofluorescence; WB, western blotting; IP, immunoprecipitation). Against mouse VE-cadherin: polyclonal antibody (pAb) C5 (prepared in-house (Gotsch et al, 1997)) (IP); pAb VE42 (prepared in-house (Broermann et al, 2011)) (IP, IF);

pAb AF1002 (WB) and pAb AF938 (IF) (R&D Systems); monoclonal antibody (mAb) to mouse PECAM-1 (1G5.1; prepared in-house (Wegmann et al, 2006)) (IF); mAb to β-catenin (610154, BD Biosciences) (WB, IF); mAb to human γ-catenin/plakoglobin (610254, BD Biosciences) (WB); mAb to actin (612656, BD Biosciences) (WB) and mAb to α-tubulin (T6074, Sigma-Aldrich) (WB). Rabbit polyclonal antibodies against plakoglobin were raised against a peptide corresponding to the C-terminal domain of murine plakoglobin (amino acids 731–745, DGLRPPYP-TADHMLA; identical to human sequence) with an additional N-terminal cysteine for coupling. Immunization of rabbits was done using a described method (Ebnet et al, 2000). Affinity purification of antibodies was performed with the synthetic peptide covalently coupled to SulfoLink Coupling Resin (Thermo Fisher Scientific, MA, USA) (IF). Secondary antibodies were as follows: Alexa Fluor 405, 488, and 647-coupled secondary antibodies were purchased from Invitrogen (CA, USA) (IF). IRDye 680RD- and IRDye 800CW coupled secondary antibodies were purchased from LI-COR Biosciences (WB). All other secondary antibodies were purchased from Jackson ImmunoResearch (WB).

## Mice

*Jup*^lox/lox^ mice, in which exon 1 of the *Jup* (plakoglobin) gene is flanked by loxP sites (Li et al, 2011), were obtained from The Jackson Laboratory. *Ctnnb1*^lox/lox^ mice, with loxP sites flanking exons 2–6 of the *Ctnnb1* (β-catenin) gene (Brault et al, 2001), were used to generate conditional knockout models. To generate endothelial-specific plakoglobin knockout mice (*Jup*^ECKO^), *Jup*^lox/lox^ mice were crossed with mice expressing Cre recombinase under the constitutively active Tie2 promoter (Tek-Cre). For conditional endothelial-specific deletion of β-catenin (*Ctnnb1*^iECKO^), *Ctnnb1*^lox/lox^ mice were crossed with mice expressing tamoxifen-inducible Cre recombinase under control of the *Pdgfb* promoter (Pdgfb-iCreER^T2^). For induction of β-catenin deletion in *Ctnnb1*^iECKO^ mice, tamoxifen (25 mg/ml in peanut oil) was administered via intraperitoneal injection (100 µl per day) for 5 consecutive days, followed by a single additional injection on day 8. Mice were used for experiments 10–11 days after the first injection. All mice were bred and maintained under specific pathogen-free conditions in the barrier facility of the Max Planck Institute for Molecular Biomedicine, with ad libitum access to food and water. All animal procedures were approved by the Landesamt für Natur, Umwelt und Verbraucherschutz Nordrhein-Westfalen, Germany (Approval numbers: 81–02.04.2020.A187 and 81–02.04.2020.A369 and 84-02.04.2017.A101).

## Generation of knock-in mice

The three VE-cadherin tension sensor knock-in mouse lines (VEC-TS, VEC-TS-NF and VEC-TS-YPet) were generated by recombinase mediated cassette exchange (RMCE) strategy in which these constructs replace the endogenous VE-cadherin in mice. The respective RMCE constructs were generated based on RMCE vector U5-3, previously created in our laboratory (Schulte et al, 2011), containing the mouse VE-cadherin cDNA, a polyA transcriptional stop cassette and a hygromycin cassette flanked with FRT sites. Using the QuikChange Site-Directed Mutagenesis Kit (Agilent Technologies) restriction sites for SalI and MluI were

introduced downstream of the codon for glycine668 of murine VE-cadherin to generate VEC-TS RMCE construct, and directly upstream of the STOP-codon of VE-cadherin to generate VEC-TS-NF RMCE construct. The cDNA of the tension sensor module YPet-FL-mCherry (Ringer et al, 2017), was PCR-amplified to include SalI and MluI restriction sites flanking the YPet-FL-mCherry sequence. The amplified tension sensor module was inserted into the respective modified murine VE-cadherin-U5-3 vector using digestion with SalI and MluI and subsequent ligation, resulting in VEC-TS and VEC-TS-NF RMCE vectors. The VEC-TS-YPet RMCE vector was generated by introducing the Y72L point mutation of mCherry into the VEC-TS construct via site-directed mutagenesis. The complete insertion cassette was flanked by incompatible loxP and lox2272 sites. Using Cre-mediated recombination, exon 2 of the endogenous *VE-cadherin* locus, which was also flanked with the same lox sites, in mouse embryonic stem (ES) cells was replaced to generate knock-in alleles. Recombinant ES cell clones were identified by PCR and Southern blot analysis and positive clones were injected into C57BL/6 blastocysts to generate chimeric mice. Genotyping was performed as previously described (Broermann et al, 2011; Schulte et al, 2011). Control mice carrying a knocked-in *VE-cadherin* cDNA (VEC-WT) were generated using the same strategy and were characterized previously by our group (Schulte et al, 2011). All knock-in mice were homozygous and were either on mixed SV129xC57Bl/6 genetic background: F1 generation (50% SV129 and C57Bl/6) or further backcrossed in the C57Bl/6 background: F6 generation (98.44% C57Bl/6).

## Intravital microscopy

Mice were injected intrascrotally with 50 ng IL-1β in 0.3 ml saline. Four hours later, mice were anaesthetized with an intraperitoneal injection of ketamine hydrochloride (125 mg/kg) and xylazine (12.5 mg/kg). Surgical preparation of cremaster muscles and intravital microscopy were carried out as described (Wessel et al, 2014; Zarbock et al, 2007). Transmigration, adhesion and rolling flux fraction of leukocytes were analyzed in 4–5 mice per group. Vessels with diameter of 20–40 µm were investigated.

## In vivo vascular permeability assay in the cremaster muscle

Vascular permeability in response to histamine was assessed in the cremaster muscle of 16–20-week-old male mice. Mice received an intravenous injection via the tail vein of 100 µl of a 1% Evans blue solution (Sigma-Aldrich) in PBS, either alone or in combination with histamine (100 µl of 8 mM solution in PBS; Sigma-Aldrich). After 15 min, mice were sacrificed, cremaster muscles were excised, and the Evans blue dye was extracted in formamide (Sigma-Aldrich) for 3 days. The amount of extravasated dye was quantified by measuring absorbance at 620 nm using a spectrophotometer (UV-1900i, Shimadzu).

## Adenoviral transduction of HUVEC

VE-cadherin tension sensor adenoviral construct (VEC-TS) was previously described (Arif et al, 2021). The VEC-TS-NF and VEC-TS-YPet constructs were essentially designed in a similar manner as

described above for the knock-in mice. The β-catenin cDNA and tension sensor construct sequences were cloned into the pENTR2B vector and transferred via an LR recombination reaction into the destination vector pAd/CMV-DEST (Gateway Technology, Invitrogen). Adenovirus was produced in HEK293A cells (Invitrogen, R70507). HUVEC were transduced with LacZ, β-catenin, VEC-TS, VEC-TS-NF or VEC-TS-YPet adenovirus and used in experiments 48 h later.

## In vitro flow-based leukocyte transmigration assay

HUVEC depleted of endogenous plakoglobin or β-catenin, or HUVEC overexpressing β-catenin or LacZ were seeded at a density of $3 \times 10^4$ cells per lane onto fibronectin-coated ibidi VI (0.4) flow slides and cultured for 48 h. Four hours prior to the assay, endothelial cells were stimulated with 5 nM TNF-α. Freshly isolated human PMNs were labeled with 2.5 μM CellTracker Deep Red (Thermo Scientific) in HBSS for 20 min at 37 °C. After washing, labeled PMNs were suspended in flow buffer (HBSS supplemented with 5 mg/ml BSA and 25 mM HEPES) at a final concentration of $0.25 \times 10^6$ cells/ml. Flow assays were performed by mounting the ibidi slides on a temperature- and $CO_2$-controlled microscope stage (37 °C, 5% $CO_2$). Slides were connected to a Harvard Apparatus syringe pump via tubing pre-equilibrated to 37 °C and flushed with warm flow buffer to eliminate air bubbles. PMNs in flow buffer were perfused over the endothelial monolayer at a shear stress of 1 dyn/cm² for 5 min, followed by a 3 min wash with PBS (containing $Ca^{2+}/Mg^{2+}$) at the same flow rate. Live-cell imaging of PMN transmigration was performed using a Zeiss LSM 880 confocal. Time-series imaging was carried out using a 633 nm laser acquiring 90 frames per video over a 212.5 × 212.5 μm field of view (512 × 512 pixels). Brightfield images were simultaneously acquired using the transmitted light photomultiplier tube (TPMT) module (Zeiss). At the end of each recording, adhered and transmigrated PMNs were quantified manually.

## Electric cell-substrate impedance sensing

HUVEC monolayer permeability was determined by measuring the electrical resistance using ECIS. Electrode arrays (8W10E; Applied BioPhysics) were pretreated with 10 mM L-cysteine (Sigma-Aldrich) for 1 h at RT, washed and subsequently coated with fibronectin (Sigma-Aldrich) for 1 h at RT. HUVEC pretreated with plakoglobin, β-catenin or control siRNA or HUVEC overexpressing β-catenin or LacZ were seeded at 40,000 cells per well and grown to confluency for 48 h. Electrical resistance was continuously measured at 37 °C at 5% $CO_2$ with ECIS Model 9600 Controller (Applied BioPhysics). Once resistance plateaued at 4000 Hz, 1 U/ml thrombin or culture medium alone was added. Electrical resistance was recorded continuously over a 2–3 h period to assess dynamic changes in barrier integrity.

## Fluoroscence recovery after photobleaching

Following transfection with either plakoglobin or β-catenin siRNA, $3 \times 10^4$ HUVEC were seeded onto fibronectin-coated 8-well ibidi chambers. After 4–6 h of attachment, cells were transduced with an adenoviral vector encoding VEC-TS-YPet and incubated for 48 h to

allow for optimal expression. The culture medium was replaced after 24 h to maintain cell viability. Live-cell imaging of YPet-positive HUVEC was carried out at 37 °C in a humidified chamber with 5% $CO_2$ using a Zeiss LSM880 confocal system (Zen 2.3 software) equipped with a Plan-Apochromat 20×/0.8 M27 objective. The imaging area was set to 42.5 μm × 18.7 μm with a pixel size of 0.08 μm. Scans were acquired in frame scan mode with 8-bit depth, a pixel dwell time of 1.02 μs, and a scan time of 138.24 ms per frame. Bleaching was performed with the 514 nm laser line at 80% power, applying the "Zoom Bleach (fast, less accurate)" mode via the ZEN FRAP module. The bleach pulse was initiated after the fifth scan out of a total of 90, and the number of bleaching iterations was set to 20. Time-lapse acquisition resumed immediately post-bleach, capturing 90 frames at 1.0-s intervals. The pinhole was set to 1.31 Airy units (2.1 μm section). For each experiment, reference and background ROIs were also defined for acquisition bleaching and background correction.

Fluorescence intensity values were extracted and analyzed using Zeiss ZEN 2 Black Edition software (version 10.0.0.910, licensed ZEN Desk, basic) with the FRAP plugin. Fluorescence intensities were background-subtracted, corrected using a reference ROI, and normalized within the software. The mobile fraction and half-time of recovery were obtained from monoexponential fits performed within ZEN.

For plotting and group analysis, normalized recovery data were exported to GraphPad Prism 10. FRAP curves for each condition were generated by averaging fluorescence values from all ROIs at each time point. A baseline offset correction was then applied using Prism's "Remove Baseline" feature, in which the value from the first post-bleach row was subtracted from all subsequent data points, ensuring that the initial post-bleach intensity aligned to zero. Final recovery curves and statistical analyses were generated in GraphPad Prism 10.

## Immunoprecipitation and immunoblotting

Murine lungs were homogenized in lysis buffer containing 10 mM sodium phosphate (pH 7.2), 150 mM NaCl, 1 mM EDTA, 0.1% SDS, 1% NP-40, 1% sodium deoxycholate, 1 mM DTT, and 2× Complete Protease Inhibitor (Roche) using a Precellys® Evolution homogenizer with CKMix 2 ml tubes (Bertin Technologies). Lysates were incubated for 3 h at 4 °C and clarified by centrifugation at $21,000 \times g$, 4 °C for 30 min. Aliquots were taken for direct immunoblot analysis. The remaining lysate was pre-cleared with Protein A Sepharose (Cytiva) coated with an isotype control antibody for 1 h at 4 °C, followed by immunoprecipitation with Protein A Sepharose coated with anti-VE-cadherin antibodies (C5 or VE42) overnight at 4 °C. Immunoprecipitates were washed five times with lysis buffer. Total HUVEC or lung lysates and immunoprecipitates were mixed with SDS sample buffer (200 mM Tris-HCl pH 6.8, 30% glycerol, 6% SDS, 0.1% bromophenol blue, 150 mM dithiothreitol) and boiled at 95 °C for 5 min. Proteins were separated by SDS–PAGE (8% gels), transferred to nitrocellulose membranes (Cytiva), and analyzed by standard western blot procedures. Chemiluminescent signals were detected using a Curix 60 film developer (Agfa) or an Odyssey Fc imaging system (LI-COR Biosciences). For fluorescence detection, an Odyssey CLx imaging system (LI-COR Biosciences) was used.

## Cremaster whole mount immunofluorescence staining

To analyze VE-cadherin expression and junctional localization in mouse venules, whole-mount staining of cremaster muscle tissue was performed. Mice were sacrificed by $CO_2$ asphyxiation. The cremaster was dissected and prefixed in situ with 4% PFA for 8 min, then removed and fixed for another 45 min at RT with 4% PFA. Next, the cremaster muscle was permeabilized and blocked in 0.5% Triton X-100, 2% ovalbumin in PBS for 2 h at RT. Thereafter, the cremaster muscles were incubated with antibodies against PECAM-1 (1G5.1) and VE-cadherin (VE42) overnight at RT followed by Alexa Fluor 647 and Alexa Fluor 405 or 488-conjugated secondary antibodies overnight at RT, respectively. Z-stack images were acquired with a Zeiss LSM 880 confocal microscope and are depicted as maximum intensity projections.

## Immunofluorescence staining in vitro

HUVEC were cultured to confluence on 8-well μ-Slides (ibidi GmbH, Gräfelfing, Germany) as described in the 'Cell culture' section. Cells were stimulated with either 100 μM histamine (Sigma-Aldrich) or 1 U/ml thrombin (Calbiochem) or left untreated with equivalent volumes of vehicle (media) as controls, each for 10 min at 37 °C. After washing once with PBS containing $Ca^{2+}$ and $Mg^{2+}$, cells were fixed with 4% paraformaldehyde (PFA) for 10 min, followed by permeabilization with 0.5% Triton X-100 for 5 min. Blocking was performed for 1 h at room temperature (RT) with 5% bovine serum albumin (BSA), and primary/secondary antibody incubations were carried out for 1 h each at RT in blocking buffer. Stained cells were covered in fluorescence mounting medium (Dako Omnis, Agilent Technologies, Santa Clara, CA, USA).

Z-stack images were acquired either on a Zeiss LSM 980 confocal microscope equipped with an Airyscan 2 detector using a Plan-Apochromat 63×/1.4 NA oil immersion objective, or on a Zeiss LSM 880 confocal microscope using a Plan-Apochromat 40×/1.2 NA water immersion objective in standard confocal mode. Alexa Fluor 488, 568, and 647 fluorophores were sequentially excited with 488 nm, 561 nm, and 633 nm lasers, respectively, with emission collected using appropriate spectral detection windows. Zeiss LSM 980 Airyscan acquisitions were processed with Airyscan Joint Deconvolution processing module in ZEN blue software (Zeiss) to improve signal-to-noise and spatial resolution; LSM 880 images were used without further deconvolution. In Figure EV3, images are depicted as maximum intensity projections.

## Time-correlated single-photon-counting fluorescence lifetime microscopy (TCSPC-FLIM)

TCSPC-FLIM experiments were conducted using a Zeiss LSM880 confocal laser scanning microscope equipped with a C-Apochromat 40×/1.20 W Korr M27 water immersion objective. Fluorescence excitation was achieved with a pulsed 485 nm diode laser (LDH-D-C-485) operating at 40 MHz. All measurements were performed under fixed conditions with laser intensity set to 67%. Emitted fluorescence was collected through a 520/35 nm bandpass filter and photon arrival time was detected using a MultiHarp 150 4 N TCSPC module with a time resolution of 80 ps. Experiments were carried out at a constant temperature of 30 °C.

The average lifetime of VEC-TS-YPet was used for each experimental day to determine $\tau_D$. FRET efficiency was calculated for each measurement, using the amplitude weighted lifetime of VEC-TS and VEC-TS-NF to determine $\tau_{DA}$, with the following calculation: $E = 1 - (\tau_{DA}/\tau_D)$.

## FLIM-FRET imaging of VE-cadherin tension sensor during leukocyte transmigration

HUVEC genetically depleted of endogenous VE-cadherin along with either plakoglobin or β-catenin, were seeded at a density of $3 \times 10^4$ cells per lane onto fibronectin-coated ibidi VI (0.4) flow slides. Following seeding, cells were transduced with VEC-TS, VEC-TS-NF, or VEC-TS-YPet constructs and incubated for 48 h to allow for expression. Four hours prior to imaging, cells were stimulated with 5 nM TNF-α. Freshly isolated human PMNs were labeled using 2.5 μM CellTracker Deep Red (Thermo Scientific, C34565) in HBSS for 20 min at 37 °C. After washing, labeled PMNs were suspended in flow buffer (HBSS supplemented with 5 mg/ml BSA and 25 mM HEPES) at a final concentration of $0.25 \times 10^6$ cells/ml. The endothelial monolayer was perfused with flow buffer containing or lacking PMNs at a shear stress of 1 dyn/cm² for 4.5 min. Cells were subsequently rinsed with PBS (containing $Ca^{2+}/Mg^{2+}$) for 1.5 min at the same flow rate and fixed in 4% PFA at 37 °C for 5 min. Fixation was terminated by washing with PBS for 3 min under continuous flow. All procedures were conducted at 37 °C in a humidified atmosphere containing 5% $CO_2$. FLIM imaging was performed using a Zeiss LSM microscope with ZEN 2.3 software and SymPhoTime 64 (v2.4, PicoQuant). Images were acquired at 3× zoom over a 70.8 × 70.8 μm area (512 × 512 pixels), with a scan speed of 16.38 μs/pixel and a pinhole size of 14 AU. Prior to lifetime acquisition, a z-stack was collected for YPet and CellTracker Deep Red using 488 nm and 633 nm lasers, respectively, at 4 μs/pixel and a pinhole of 2 AU in the same field of view. For FLIM-FRET analysis, photon data from endothelial junctions adjacent to migrating PMNs were manually selected. Lifetime fitting was performed in SymPhoTime software using a one-exponential tail fit for VEC-TS-YPet and a two-exponential model for VEC-TS and VEC-TS-NF.

## FLIM-FRET imaging of VE-cadherin tension sensor during thrombin or histamine stimulation in vitro

HUVEC lacking endogenous VE-cadherin together with either plakoglobin or β-catenin were seeded at a density of $3 \times 10^4$ cells per chamber on fibronectin-coated 8-well μ-slide chambers (ibidi). Following seeding, cells were transduced with VEC-TS, VEC-TS-NF, or VEC-TS-YPet constructs and incubated for 48 h to allow for expression. Cells were stimulated with either 100 μM histamine (Sigma-Aldrich) or 1 U/ml thrombin (Calbiochem) or left untreated for 10 min at 37 °C. Following stimulation, cells were fixed with 4% PFA in PBS (containing $Ca^{2+}/Mg^{2+}$) for 5 min at 37 °C and subsequently washed with PBS. FLIM imaging was performed and analyzed as described above.

## FLIM-FRET imaging in cremaster muscle of VE-cadherin tension sensor knock-in mice during histamine stimulation

To assess tension across VE-cadherin in vivo, whole-mounts of mouse cremaster muscles were prepared following histamine-

induced vascular permeability. Homozygous VEC-TS and VEC-TS-NF mice were injected via the tail vein with histamine (100 µl of 8 mM solution in PBS; Sigma-Aldrich), co-administered with crimson-labeled 20 nm FluoSpheres fluorescent microspheres (0.5% solid in 100 µl PBS; Thermo Scientific) or microspheres alone, for 3 min to label sites of vascular leakage. Mice were then euthanized, and the cremaster muscles were dissected and left in situ for pre-fixation with 4% PFA in PBS for 8 min. Tissues were subsequently excised and post-fixed in 4% PFA for an additional 45 min, followed by PBS washes to remove residual fixative. Samples were mounted in DAKO mounting medium on glass slides for imaging. FLIM imaging was performed and analyzed as described above for HUVEC, with the exception that in cremaster whole mounts FLIM measurements were acquired as image stacks and analyzed using the Multi-frame FLIM mode.

### Statistical analysis and software

Total sample numbers were determined on the basis of previous studies with transgenic mouse models. Statistical analysis was performed using GraphPad Prism 10 software (GraphPad Software Inc.). Data are shown as mean ± standard error of the mean (SEM). Statistical significance was analyzed using an unpaired two-tailed t-test, Mann–Whitney test, one-way ANOVA or two-way ANOVA. Significance thresholds were defined as $p < 0.05$ (*), $p < 0.01$ (**), $p < 0.001$ (***), and $p < 0.0001$ (****). Confocal microscopy data were acquired using Zeiss confocal microscopes with Zen 2.3 SP1 Black (64-bit) software, and maximum intensity projections of immunofluorescence images were generated using Fiji-ImageJ. FRAP analysis was performed using the FRAP plugin in Zeiss ZEN 2 Black Edition software (version 10.0.0.910, licensed ZEN Desk, basic). Immunoblot signal quantification was performed using Image Studio (LI-COR Biosciences) or Fiji-ImageJ. FLIM-FRET measurements and analyses were conducted using SymPhoTime 64 software.

## Data availability

This study includes no data deposited in external repositories.

The source data of this paper are collected in the following database record: biostudies:S-SCDT-10_1038-S44318-026-00732-0.

## Peer review information

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

## Acknowledgements

We thank Marika Meyer zu Brickwedde for excellent technical assistance. We are very grateful to the staff of the following core facilities of the Max Planck Institute for Molecular Biomedicine: the biooptics core facility for excellent help in image acquisition and evaluation, the transgenic core facility for great

help in generating the genetically modified mouse lines and the animal care facility for thorough and reliable mouse breeding. This work was supported by funds from the Max Planck Society (DV) by funds from the CIM-IMPRS graduate school and grants from the Deutsche Forschungsgemeinschaft (SFB1348, B01) to DV and from the Medical Scientist Program of the Else Kröner Fresenius Foundation to RIS. Open Access funding enabled and organized by ProjektDEAL.

## Author contributions

**Neha Uttekar**: Data curation; Formal analysis; Validation; Methodology; Writing—review and editing. **Annette Artz**: Data curation; Formal analysis; Validation. **Vallari Ghanekar**: Data curation; Formal analysis; Validation; Writing—review and editing. **Pragya Kaul**: Data curation; Formal analysis; Validation. **Jessica Heinrichs**: Data curation; Formal analysis; Validation. **Rebekka I Stegmeyer**: Methodology. **Gizem Gülevin Takir**: Methodology. **Astrid F Nottebaum**: Data curation; Formal analysis; Validation; Methodology; Writing—review and editing. **Dietmar Vestweber**: Conceptualization; Supervision; Investigation; Writing—original draft; Project administration; Writing—review and editing.

Source data underlying figure panels in this paper may have individual authorship assigned. Where available, figure panel/source data authorship is listed in the following database record: biostudies:S-SCDT-10_1038-S44318-026-00732-0.

## Funding

## Disclosure and competing interests statement

The authors declare no competing interests.

# Expanded View Figures

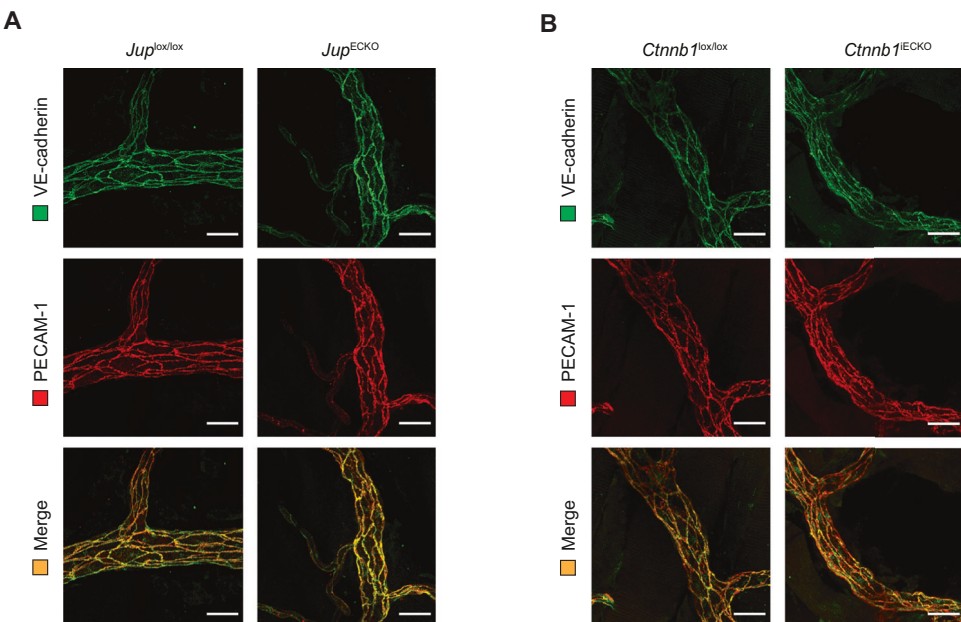

**Figure EV1. Loss of plakoglobin or β-catenin does not affect VE-cadherin expression in vivo.**

(A, B) Whole-mount immunostaining of cremaster muscle from *Jup*^lox/lox and *Jup*^ECKO mice (A) or *Ctnnb1*^lox/lox and *Ctnnb1*^iECKO mice (B), with anti-VE-cadherin and anti-PECAM-1 antibodies, presented as maximum intensity projections of Z-stacks. Scale bars, 25 μm.

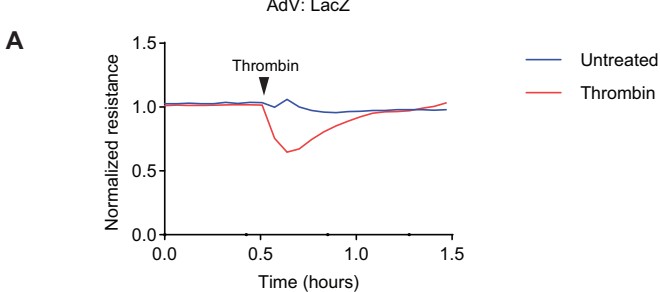

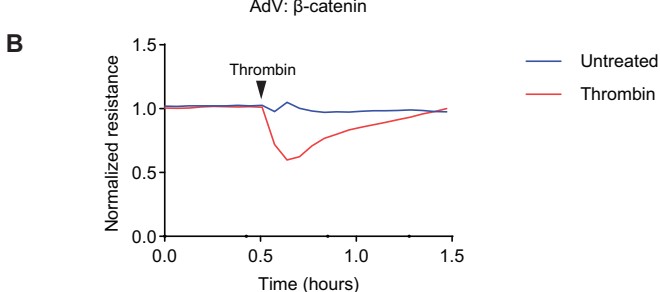

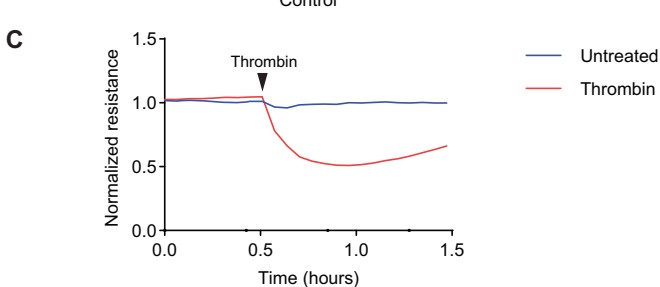

**Figure EV2. Electrical resistance measurements of HUVEC exposed to adenovirus vectors.**

(A–C) HUVEC were transduced with LacZ (A), or β-catenin (B) adenovirus, or left untransduced as control (C), and grown to confluency on fibronectin-coated electrode arrays. Monolayers were subsequently stimulated with 1 U/mL thrombin (red lines) or left untreated (blue lines), and transendothelial electrical resistance was recorded over time by ECIS. Representative graphs show resistance after normalization to the resistance of HUVEC monolayers before thrombin treatment.

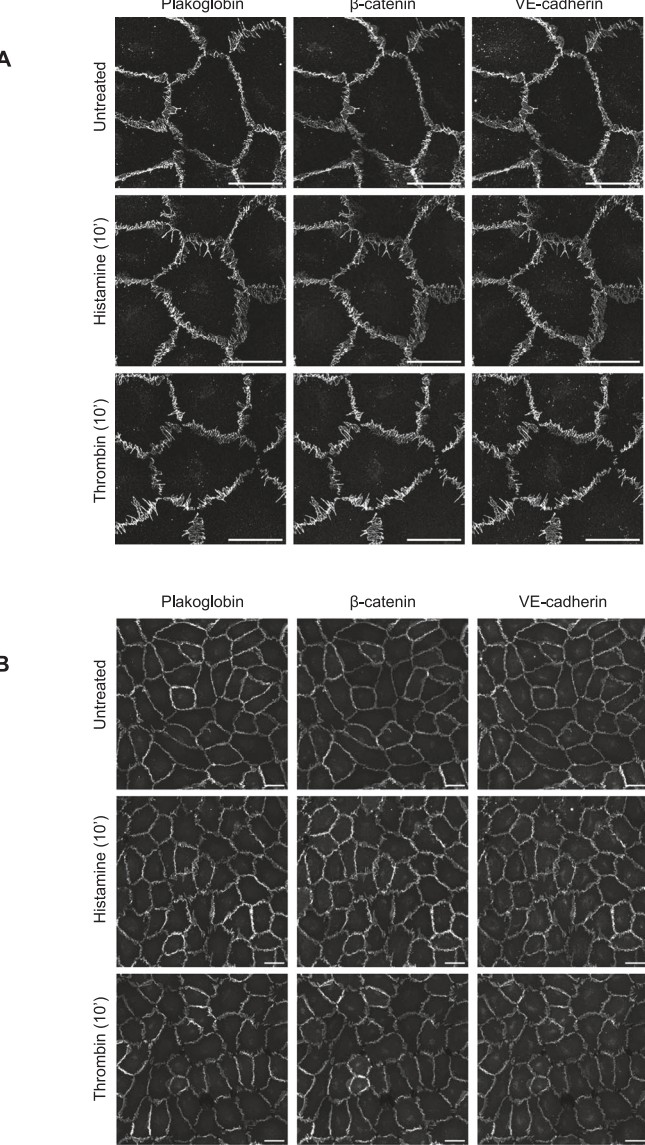

**Figure EV3. Plakoglobin or β-catenin distribution showed no major differences within inflamed and control endothelial monolayers.**

(A, B) HUVEC were treated with histamine (100 μM, Sigma-Aldrich) or thrombin (1 U/ml, CalBiochem) for 10 min, or left untreated with equivalent volumes of vehicle (media) as controls. Cells were subsequently fixed, permeabilized and stained for plakoglobin, β-catenin, and VE-cadherin, all visualized in grayscale to ensure unbiased comparative analysis. Higher magnification images (63X) were acquired using a Zeiss LSM 980 confocal microscope equipped with an Airyscan detector (A), while lower magnification images (40X) were captured on a Zeiss LSM 880 confocal microscope (B). Scale bars, 25 μm.

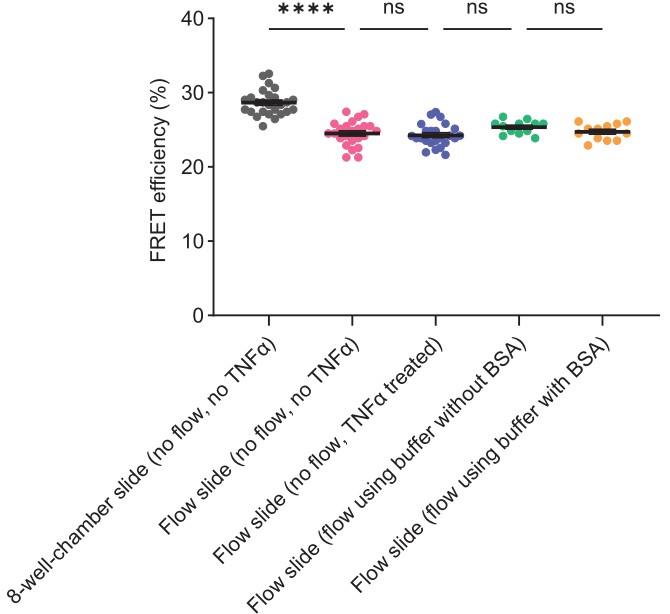

**Figure EV4. Variability in FRET efficiency (%) of VEC-TS-NF construct.**

FRET efficiency at endothelial junctions was quantified in HUVEC expressing VEC-TS-NF and cultured in ibidi 8-well chambers or flow chamber slides under distinct conditions, including TNFα stimulation (4 h) and exposure to flow (1 dyn/cm²) using flow buffer with or without BSA. Cells were fixed with 4% PFA, washed with PBS, and analyzed by FLIM. Data information: Graph shows mean ± SEM from ($n = 26, 24, 24, 12, 12$) measurements. ****$P < 0.0001$, one-way ANOVA. ns, not significant.

