## [Peer Review File · The EMBO Journal]

Plakoglobin transmits tension across VE-cadherin for vascular leak formation and leukocyte diapedesis

Neha Uttekar, Annette Artz, Vallari Ghanekar, Pragya Kaul, Jessica Heinrichs, Rebekka Stegmeyer, Gizem Gülevin Takir, Astrid Nottebaum, and Dietmar Vestweber

Corresponding author: Dietmar Vestweber (vestweb@mpi-muenster.mpg.de)

Review Timeline:

Submission Date:	8th Jul 25
Editorial Decision:	19th Aug 25
Revision Received:	8th Dec 25
Editorial Decision:	22nd Jan 26
Revision Received:	2nd Feb 26
Accepted:	10th Feb 26

Editor: Ieva Gailite

Transaction Report:

Dear Dietmar,

Thank you for submitting your manuscript for consideration by the EMBO Journal. We have now received comments from a full set of reviewers, which are included below for your information.

As you will see, all reviewers are generally positive in their evaluation and appreciate the contribution of the study to the research field. They also indicate a number of reasonable and overlapping concerns that would need to be addressed before they can support publication here, including a potential compensatory mechanism via beta-catenin upregulation upon plakoglobin deficiency (reviewer #1, point 3; reviewer #2 point 2), quantification of the impact of plakoglobin-mediated tension on VE-cadherin localisation (reviewer #1, point 1; reviewer #3, point 2), in addition to other technical requests. Based on these positive assessments, I invite you to submit a revised manuscript in response to the comments by all reviewers. I would also be happy to discuss the revision in more detail via email or phone/videoconferencing.

We generally allow three months as standard revision time, which can be extended to six months in the case of major revisions. Should you foresee a problem in meeting this deadline, please let us know in advance to discuss an extension.

As a matter of policy, competing manuscripts published during this period will not negatively impact on our assessment of the conceptual advance presented by your study. However, please contact me as soon as possible upon publication of any related work to discuss the appropriate course of action.

When preparing your letter of response to the referees' comments, please bear in mind that this will form part of the Review Process File and will therefore be available online to the community. For more details on our Transparent Editorial Process, please visit our website: <https://www.embopress.org/page/journal/14602075/authorguide#transparentprocess>. Please also see the attached instructions for further guidelines on preparation of the revised manuscript.

Please feel free to contact me if you have any further questions regarding the revision. Thank you for the opportunity to consider your work for publication. I look forward to your revision.

With best wishes,

Ieva

- a point-by-point response to the referees' comments, with a detailed description of the changes made (as a word file).
- a word file of the manuscript text.
- individual production quality figure files (one file per figure)
- a complete author checklist, which you can download from our author guidelines (<https://www.embopress.org/page/journal/14602075/authorguide>).

- Expanded View files (replacing Supplementary Information)

- a Reagents and Tools Table as part of the Methods section, which can be downloaded from our author guidelines

(<https://www.embopress.org/page/journal/14602075/authorguide#structuredmethods>)

We realize that it is difficult to revise to a specific deadline. In the interest of protecting the conceptual advance provided by the work, we recommend a revision within 3 months (17th Nov 2025). Please discuss the revision progress ahead of this time with the editor if you require more time to complete the revisions.

Referee #1:

In this manuscript by Uttekar et al the authors investigated the contributions of beta-catenin and plakoglobin (also known as gamma-catenin) to VE-cadherin in endothelial cells.

The work is of high impact for its ability to discern the differences in these two catenins with regards to endothelial adherens junctions. In addition to the careful analysis of each catenin, the work is also significant for in vivo analysis of the effects of histamine on forces at the cell-cell adhesion (a first for the field, and thus this is very impactful beyond these two catenins). I also think the use of the VE-cadherin tension sensor in mice was a very cutting edge approach and may be of broad interest to others who may also be considering the use of biosensors in mice.

I am very supportive of this manuscript for EMBO.

I have some concerns I hope the authors would be able to address in a revised manuscript:

1. The no force (NF) sensor appears to have some variability in the FRET efficiency across the figures. Notably in figure 5B and 5C the NF FRET appears possibly lower than the TS. This is confusing, since the VE-cadherin TS should report similar FRET (indicating no force) or lower FRET (indicating tensile force). Can the authors address this potential discrepancy?
 2. Do the authors have any data to show on how each catenin affects VE-cadherin at cell-cell adhesions? For example immunostaining to show if the morphology, intensity, etc. are different? I see this was provided in some staining in Figure EV1 (a supplemental figure), but I wondered if there were some higher magnification images to better highlight this? OR some analysis in vitro? For example, some simple experiments with FRAP could help understand if the dynamics of VE-cadherin are altered depending on the specific catenin.
 3. The authors make the conclusion that plakoglobin is the major agent for change during histamine stimulation (including increasing force). However, it is possible that the converse is actually more important, that somehow beta-catenin recruitment is activating some signaling pathway that limits histamine-induced responses or creating a highly stable cell-cell adhesion that prevents leukocyte diapedesis. Given the upregulation of beta-catenin in the plakoglobin KD experiments, it is hard to discern if the loss of plakoglobin or the increased recruitment of beta-catenin is the more important factor. Did the authors attempt to do any beta-catenin overexpression experiments (in vitro) to see if beta-catenin over expression recapitulates (or does not recapitulate) the leukocyte transmigration or histamine changes?
 4. Are the authors able to offer any speculation (or cite studies) on how:
A) plakoglobin vs beta catenin junctional recruitment may change during histamine stimulation and/or leukocyte transmigration?
B) what might drive plakoglobin vs beta-catenin interactions with VE-cadherin?
C) what might be the variations in the downstream binding partners of AJs with beta-catenin vs plakoglobin?
- I think it would be unreasonable to ask the authors to do any experiments to address these questions as part of this manuscript, but it would be helpful to somehow discuss one or more of these points in the discussion.

Referee #2:

VE Cadherin, an endothelial cell (EC)-specific protein, anchors the cell-cell junctions to the cytoskeleton through catenins, thus regulating junctional tension and permeability. -catenin and plakoglobin (γ -catenin) have been proposed to both mediate this process in distinct/co-existing complexes. The study by Uttekar et al., aims to directly investigate and compare the functional

roles of β -catenin and plakoglobin in vascular leakage and leukocyte diapedesis. Through cultured EC siRNA knockdown in vitro studies and EC-specific gene deletion in vivo studies, the authors provide evidence for plakoglobin, but not β -catenin, in supporting junctional permeability in response to inflammatory mediators and cytokine-induced leukocyte transendothelial migration through mouse cremasteric venules. Furthermore, using a novel VE-Cadherin tension sensor and a knock in mouse model, they provide evidence to suggest that plakoglobin is dominantly required for mediating tension placed on VE-cadherin, presumably through cytoskeletal linkage. Collectively, through use of multiple novel tools, well-planned and controlled studies, the findings describe distinct functionality of these homologues. The investigation would however benefit from more molecular detail.

Major concerns:

1. Whilst providing elegant data on differing relevance of β -catenin and plakoglobin in regulation of vascular junctions, the study is weak on mechanistic detail. For example, the potential role of compensatory overexpression of β -catenin following plakoglobin KD requires attention (e.g. Figure 1B). This could be addressed using in vitro over-expression models to test whether increased expression of β -catenin is a component of the effects seen on junctional functions under conditions of plakoglobin KD. Furthermore, the manuscript would benefit from additional molecular insights into how plakoglobin is mediating these effects. This is the major shortcoming of this otherwise well designed and executed study.

2. Whilst it is appreciated that the generation of an EC-specific β -catenin KO mouse is not possible: a KD efficiency of 40% may not be enough to observe a biological effect, and a potential reason why no effect is observed. Did the authors quantify β -catenin protein levels in cremasteric venular ECs? Whilst in vitro studies provide some support for the authors in vivo conclusions, the data in Figure 1D suggests that with increased experimental repeats the in vitro adhesion data under conditions of siRNA β -catenin KD could be significant.

Minor concerns:

1. Reduced leukocyte TEM is often linked with increased adhesion. Please comment on why this is not seen under conditions of plakoglobin KD.

2. For continuity and transparency, data points should be shown in Figures 1 C-D and 2 E-J.

3. Figure 8A: The images are very clear however quantification would have enhanced the rigour of the data. Panels C/D: Statistical analysis on n=2 data?

Referee #3:

Vascular endothelial (VE)-cadherin serves as a pivotal component in the maintenance of endothelial junctional integrity, thereby governing crucial physiological processes such as inflammation-induced vascular permeability and leukocyte transendothelial migration. The adhesive properties of VE-cadherin are regulated through its association with intracellular binding partners, most notably β -catenin. This β -catenin, through its linkage by α -catenin, integrates VE-cadherin with the actin cytoskeleton, a structural organization fundamental to the stability and dynamic regulation of endothelial junctions. Notably, plakoglobin (also known as γ -catenin), a close structural homolog of β -catenin, is capable of substituting β -catenin in the VE-cadherin-catenin complex, and both types of complexes can co-exist within endothelial cells. However, the exact role of plakoglobin in the regulation of endothelial cell-cell contacts is not clear.

To elucidate the respective roles of β -catenin and plakoglobin in the modulation of endothelial junctions, the present study employed a combination of gene silencing approaches in vitro and conditional, endothelium-specific gene ablation strategies in murine models in vivo. The findings revealed a striking divergence in the functional relevance of these two catenins: while β -catenin appeared to be largely dispensable, plakoglobin proved to be indispensable for both leukocyte transmigration across the endothelium and for the enhancement of vascular permeability triggered by inflammatory agents. This finding is exciting as no clear role for plakoglobin has been described and this study addresses this for the first time.

Mechanistically, the study showed that plakoglobin is of importance for the generation of mechanical tension across VE-cadherin during leukocyte transmigration and upon stimulation with inflammatory mediators, processes that β -catenin fails to recapitulate. This conclusion was supported by the generation of transgenic mice expressing a VE-cadherin-based tension sensor, which provided in vivo evidence for the essential role of plakoglobin in mediating histamine-induced tension across endothelial junctions.

It was an enjoyable read and I applaud the authors for their work. This study substantially advances our understanding of vascular biology, elucidating a yet unrecognized specificity in the molecular machinery that controls endothelial barrier dynamics under inflammatory conditions. The findings underscore plakoglobin as a critical regulator of endothelial function with potential implications for therapeutic targeting of vascular dysfunction in inflammation. The work is thorough and conceptually robust, with

only a few suggestions to further enhance the study's impact. I have listed them below.

1. The manuscript would benefit from the inclusion of immunostaining data comparing plakoglobin and β -catenin distribution within inflamed versus control endothelial monolayers. While the discussion mentions no differences were observed, clarification on whether these experiments were performed would be valuable. Additionally, the authors might consider discussing potential heterogeneity in catenin distribution at the monolayer or single-cell level, as well as the implications for localized endothelial junction remodeling or transmigration "hotspots." If technical constraints such as antibody specificity preclude this, elaboration in the discussion is encouraged.

2. Although plakoglobin knockdown markedly reduces leukocyte diapedesis, the mechanistic link between VE-cadherin tension modulation and leukocyte transendothelial migration (TEM) merits further clarification. Does plakoglobin-mediated tension facilitate the dissociation of VE-cadherin dimers or promote local internalization of VE-cadherin complexes? Moreover, how might these processes influence endothelial permeability during TEM? Given prior evidence from this group and others indicating that leukocyte TEM and vascular permeability can be uncoupled, additional discussion on this point would strengthen the manuscript.

3. In addressing the *in vivo* tension measurements, the authors note the limitation in assessing neutrophil transmigration-induced VE-cadherin tension. It may be pertinent to reference prior phospho-myosin light chain (phospho-MLC) staining studies in inflamed cremaster muscle tissue (Heemskerk et al., 2016, Figure 6F), which provide evidence of actomyosin-generated tension during diapedesis. Incorporating this information would contextualize and reinforce the mechanistic narrative.

Collectively, these data denote an unappreciated specificity in the molecular machinery underlying endothelial barrier regulation, demonstrating that plakoglobin, but not β -catenin, is required for leukocyte diapedesis, inflammatory induction of vascular permeability, and the transmission of mechanical forces across VE-cadherin. The study thus highlights plakoglobin as a critical determinant of endothelial function during inflammatory responses, with potential implications for targeting vascular dysfunction in pathological inflammation.

Point by point response to reviewer's comments

We thank the reviewers for their positive, helpful and very constructive comments that we have addressed below as follows:

Referee #1:

In this manuscript by Uttekar et al the authors investigated the contributions of beta-catenin and plakoglobin (also known as gamma-catenin) to VE-cadherin in endothelial cells. The work is of high impact for its ability to discern the differences in these two catenins with regards to endothelial adherens junctions. In addition to the careful analysis of each catenin, the work is also significant for in vivo analysis of the effects of histamine on forces at the cell-cell adhesion (a first for the field, and thus this is very impactful beyond these two catenins). I also think the use of the VE-cadherin tension sensor in mice was a very cutting edge approach and may be of broad interest to others who may also be considering the use of biosensors in mice.

I am very supportive of this manuscript for EMBO.

I have some concerns I hope the authors would be able to address in a revised manuscript:

1. The no force (NF) sensor appears to have some variability in the FRET efficiency across the figures. Notably in figure 5B and 5C the NF FRET appears possibly lower than the TS. This is confusing, since the VE-cadherin TS should report similar FRET (indicating no force) or lower FRET (indicating tensile force). Can the authors address this potential discrepancy?

We agree with the reviewer, the no force (NF)-construct shows some variation in FRET efficiency in different types of experiments (difference between new Fig. 7 and new Fig. 8). We have repeated these experiments now for the NF-construct and report in a new results chapter and show in the new figure EV4 that FRET efficiency of the NF construct is indeed lower in HUVEC grown in fibronectin-coated flow chambers (ibidi) than in cells cultured in fibronectin-coated 8-well chamber slides (ibidi). Since FLIM measurements with cells grown in flow chambers were done with TNF- α treated cells and after incubation under flow with Ca²⁺/Mg²⁺-buffer containing BSA (old Fig. 5, now Fig. 7), we also compared conditions with and without TNF- α and with flow buffer with or without BSA (New figure EV4). We found that neither the different medium/buffer conditions nor TNF- α were the reason for reduced FRET efficiency of the NF construct in HUVEC in flow chambers.

We do not know the underlying reason. It is possible that oxygen supply of the cells may be different between flow chambers and the 8-well-slides and the two chromophores in the NF construct might be more sensitive to these changes than the chromophores in the force-sensitive construct. Since the chromophore-pair is located in a different position in the two

constructs, the sensor could be in a slightly different molecular environment, which may cause this different behavior.

Interestingly, we made similar observations for the other NF construct that we used in a previous paper (Arif et al., EMBO J. 2021, PMID: 33604918, compare figure 7 with figure 8). This older construct was designed the same way as the original NF construct that was used in the Schwartz lab. In this NF construct, the sensor was in the same position of the cytoplasmic tail of VE-cadherin as in the force-sensing construct, whereas the C-terminus of VE-cadherin containing the α -catenin binding site had been deleted. Although the two NF-constructs carry the sensor in a different position within the cytoplasmic domain of VE-cadherin, they have in common that the sensor is located at the very C-terminus of a fusion protein. We speculate that this might be the reason why the two NF-constructs both are sensitive to changes in culture conditions (flow chamber vs. 8-well-slide), whereas the force-sensing construct is not. Whatever may be the correct explanation for this unexpected behavior of the two NF-constructs, we think they are still valid as no-force control constructs and both constructs are not sensitive to signaling induced by thrombin, histamine or neutrophils.

2. Do the authors have any data to show on how each catenin affects VE-cadherin at cell-cell adhesions? For example, immunostaining to show if the morphology, intensity, etc. are different? I see this was provided in some staining in Figure EV1 (a supplemental figure), but I wondered if there were some higher magnification images to better highlight this? OR some analysis in vitro? For example, some simple experiments with FRAP could help understand if the dynamics of VE-cadherin are altered depending on the specific catenin.

We have now followed the suggestion of the reviewer and tested whether siRNA for plakoglobin or β -catenin would affect the dynamics of VE-cadherin at cell contacts of HUVEC. Based on FRAP experiments we now show in the new Fig. 6, neither half time recovery nor mobile fraction of VE-cadherin were differently affected by either plakoglobin or β -catenin siRNA treatment. In addition, we document in the new figure EV3, (based on polyclonal antibodies we raised and affinity purified against murine plakoglobin) that the distribution of plakoglobin and β -catenin is indistinguishable in HUVEC as shown in different magnifications.

3. The authors make the conclusion that plakoglobin is the major agent for change during histamine stimulation (including increasing force). However, it is possible that the converse is actually more important, that somehow beta-catenin recruitment is activating some signaling pathway that limits histamine-induced responses or creating a highly stable cell-cell adhesion that prevents leukocyte diapedesis. Given the upregulation of beta-catenin in the plakoglobin KD experiments, it is hard to discern if the loss of plakoglobin or the increased recruitment of beta-catenin is the more important factor. Did the authors attempt to do any beta-catenin overexpression experiments (in vitro) to see if beta-catenin over expression recapitulates (or does not recapitulate) the leukocyte transmigration or histamine changes?

This is indeed a very good point. We have overexpressed β -catenin in HUVEC by adenovirus transduction and tested whether this would affect leukocyte diapedesis or thrombin-induced interference with endothelial cell contact integrity. As shown in the new Fig. 5, we

found no such effect. Overexpression of β -catenin did not affect diapedesis of neutrophils (Fig. 5 D) and did not alter the effect of thrombin on electrical resistance to alternating current (ECIS) of HUVEC monolayers (Fig. 5 F-H). As a negative control we used an adenovirus vector expressing LacZ. Thus, we confirmed, that silencing of plakoglobin interferes with leukocyte diapedesis and the thrombin effect on junction integrity due to the lack of plakoglobin and not due to the increase in β -catenin expression.

In the course of these new experiments, we found, unexpectedly, that adenovirus-based transduction of HUVEC (independent of whether β -catenin or LacZ was expressed) accelerated the recovery of electrical resistance. This became apparent when the effect was compared with HUVEC which were not transduced with adenovirus vectors (new Fig. EV2). Although we do not know the reason for this adenovirus-based effect, it does not affect our conclusion mentioned above.

4. Are the authors able to offer any speculation (or cite studies) on how:

A) plakoglobin vs beta catenin junctional recruitment may change during histamine stimulation and/or leukocyte transmigration?

B) what might drive plakoglobin vs beta-catenin interactions with VE-cadherin?

C) what might be the variations in the downstream binding partners of AJs with beta-catenin vs plakoglobin?

I think it would be unreasonable to ask the authors to do any experiments to address these questions as part of this manuscript, but it would be helpful to somehow discuss one or more of these points in the discussion.

We agree, these points are indeed interesting and important. We have mentioned and discussed these points now on pages 17/18. However, unfortunately, there is not yet much known in the literature about how plakoglobin and β -catenin function at endothelial junctions.

Referee #2:

VE Cadherin, an endothelial cell (EC)-specific protein, anchors the cell-cell junctions to the cytoskeleton through catenins, thus regulating junctional tension and permeability. β -catenin and plakoglobin (γ -catenin) have been proposed to both mediate this process in distinct/co-existing complexes. The study by Uttekar et al., aims to directly investigate and compare the functional roles of β -catenin and plakoglobin in vascular leakage and leukocyte diapedesis. Through cultured EC siRNA knockdown in vitro studies and EC-specific gene deletion in vivo studies, the authors provide evidence for plakoglobin, but not β -catenin, in supporting junctional permeability in response to inflammatory mediators and cytokine-induced leukocyte transendothelial migration through mouse cremasteric venules. Furthermore, using a novel VE-Cadherin tension sensor and a knock in mouse model, they provide evidence to suggest that plakoglobin is dominantly required for mediating tension placed on VE-cadherin, presumably through cytoskeletal linkage. Collectively, through use of multiple novel tools, well-planned and controlled studies, the findings describe distinct functionality of these homologues. The investigation would however benefit from more molecular detail.

Major concerns:

1. Whilst providing elegant data on differing relevance of β -catenin and plakoglobin in regulation of vascular junctions, the study is weak on mechanistic detail. For example, the potential role of compensatory overexpression of β -catenin following plakoglobin KD requires attention (e.g. Figure 1B). This could be addressed using in vitro over-expression models to test whether increased expression of β -catenin is a component of the effects seen on junctional functions under conditions of plakoglobin KD. Furthermore, the manuscript would benefit from additional molecular insights into how plakoglobin is mediating these effects. This is the major shortcoming of this otherwise well designed and executed study.

We have now followed the suggestion of the reviewer and tested whether overexpression of β -catenin would affect leukocyte diapedesis or thrombin-induced interference with endothelial cell contact integrity. We found no such effect as we already described in our response to point 3 of reviewer 1 (see results in new Fig. 5). Overexpression of β -catenin did not affect diapedesis of neutrophils (Fig. 5 D) and did not alter the effect of thrombin on electrical resistance to alternating current (ECIS) of HUVEC monolayers (Fig. 5 F-H). As a negative control we used an adenovirus vector expressing LacZ. Thus, we confirmed, that silencing of plakoglobin interferes with leukocyte diapedesis and the thrombin effect on junction integrity due to the lack of plakoglobin and not due to the increase in β -catenin expression.

In the course of these new experiments, we found, unexpectedly, that adenovirus-based transduction of HUVEC (independent of whether β -catenin or LacZ was expressed) accelerated the recovery of electrical resistance. This became apparent when the effect was compared with HUVEC which were not transduced with adenovirus vectors (new Fig. EV2). Although we do not know the reason for this adenovirus-based effect, it does not affect our conclusion mentioned above.

2. Whilst it is appreciated that the generation of an EC-specific β -catenin KO mouse is not possible: a KD efficiency of 40% may not be enough to observe a biological effect, and a potential reason why no effect is observed. Did the authors quantify β -catenin protein levels in cremasteric venular ECs? Whilst in vitro studies provide some support for the authors in vivo conclusions, the data in Figure 1D suggests that with increased experimental repeats the in vitro adhesion data under conditions of siRNA β -catenin KD could be significant.

We appreciate the suggestion of the reviewer, yet we could not quantify β -catenin protein levels in cremaster venular EC since β -catenin is expressed in so many cell types and it is technically difficult to distinguish EC staining from the staining of other cells within close vicinity.

As suggested by the reviewer, we repeated the experiments shown in Fig. 1D and could confirm that β -catenin siRNA had indeed no significant effect on PMN adherence to endothelial cells in transmigration experiments under flow (see new part D of Fig. 1).

Minor concerns:

1. Reduced leukocyte TEM is often linked with increased adhesion. Please comment on why this is not seen under conditions of plakoglobin KD.

We agree, it has often been observed *in vivo* that inhibition of the diapedesis process can indirectly cause an increase in PMN adhesion to inflamed venular endothelium. This is often a transient effect, since neutrophils which are hindered to transmigrate do eventually detach and get washed away with the blood stream. Yet, for a while they do indeed stay attached and accumulate for some time. In contrast to such *in vivo* settings, in our *in vitro* transmigration assays under flow, we wash the flow chamber at the end of the assays with $\text{Ca}^{2+}/\text{Mg}^{2+}$ -containing PBS, in order to remove non-adherent cells. It may be possible that some of the cells which were hindered to transmigrate can get washed away under such conditions.

2. For continuity and transparency, data points should be shown in Figures 1 C-D and 2 E-J.

We have changed the display of our data in figures 1C-D and 2 E-J accordingly.

3. Figure 8A: The images are very clear however quantification would have enhanced the rigour of the data. Panels C/D: Statistical analysis on n=2 data?

We have now quantified the accumulation of beads in mice expressing either the VEC-TS construct or the VEC-TS-NF construct and show the result incorporated in new Fig. 10B. We found no difference in bead accumulation between the two mouse strains.

With respect to the statistical analysis of panels D/E of new Fig. 10 (panels C/D of old Fig. 8), we had originally analyzed 2 mice for every genotype and condition in old Fig. 8C (32 measurements from 16 vessels for each genotype and condition) and old figure 8D (20 measurements from 10 vessels for each genotype and condition). We have now increased the number of mice from 2 to 3 for each genotype and condition in Fig. 10E. Thus, in Fig. 10D, we show results from 32 measurements per group pooled from two independent experiments, and in Fig. 10E we show n = 25, 30, 18, or 16 measurements for the different genotypes and conditions, pooled from three independent experiments (one mouse per experiment in D, E).

Referee #3:

Vascular endothelial (VE)-cadherin serves as a pivotal component in the maintenance of endothelial junctional integrity, thereby governing crucial physiological processes such as inflammation-induced vascular permeability and leukocyte transendothelial migration. The adhesive properties of VE-cadherin are regulated through its association with intracellular binding partners, most notably β -catenin. This β -catenin, through its linkage by α -catenin, integrates VE-cadherin with the actin cytoskeleton, a structural organization fundamental to

the stability and dynamic regulation of endothelial junctions. Notably, plakoglobin (also known as γ -catenin), a close structural homolog of β -catenin, is capable of substituting β -catenin in the VE-cadherin-catenin complex, and both types of complexes can co-exist within endothelial cells. However, the exact role of plakoglobin in the regulation of endothelial cell-cell contacts is not clear.

To elucidate the respective roles of β -catenin and plakoglobin in the modulation of endothelial junctions, the present study employed a combination of gene silencing approaches in vitro and conditional, endothelium-specific gene ablation strategies in murine models in vivo. The findings revealed a striking divergence in the functional relevance of these two catenins: while β -catenin appeared to be largely dispensable, plakoglobin proved to be indispensable for both leukocyte transmigration across the endothelium and for the enhancement of vascular permeability triggered by inflammatory agents. This finding is exciting as no clear role for plakoglobin has been described and this study addresses this for the first time.

Mechanistically, the study showed that plakoglobin is of importance for the generation of mechanical tension across VE-cadherin during leukocyte transmigration and upon stimulation with inflammatory mediators, processes that β -catenin fails to recapitulate. This conclusion was supported by the generation of transgenic mice expressing a VE-cadherin-based tension sensor, which provided in vivo evidence for the essential role of plakoglobin in mediating histamine-induced tension across endothelial junctions.

It was an enjoyable read and I applaud the authors for their work. This study substantially advances our understanding of vascular biology, elucidating a yet unrecognized specificity in the molecular machinery that controls endothelial barrier dynamics under inflammatory conditions. The findings underscore plakoglobin as a critical regulator of endothelial function with potential implications for therapeutic targeting of vascular dysfunction in inflammation. The work is thorough and conceptually robust, with only a few suggestions to further enhance the study's impact. I have listed them below.

1. The manuscript would benefit from the inclusion of immunostaining data comparing plakoglobin and β -catenin distribution within inflamed versus control endothelial monolayers. While the discussion mentions no differences were observed, clarification on whether these experiments were performed would be valuable. Additionally, the authors might consider discussing potential heterogeneity in catenin distribution at the monolayer or single-cell level, as well as the implications for localized endothelial junction remodeling or transmigration "hotspots." If technical constraints such as antibody specificity preclude this, elaboration in the discussion is encouraged.

We have now added images depicting immunofluorescence staining of HUVEC for plakoglobin, β -catenin and VE-cadherin. The cells were either untreated or treated with histamine or thrombin. Results are shown at very high resolution (achieved with an LSM980, Airyscan, 63x objective) and images obtained with a 40x objective, to have a larger overview. Unfortunately, we did not see significant differences in the distribution of plakoglobin and β -catenin at maximal resolution (Fig. EV3). Images displaying more cells (40x objective) and therefore allowing a better overview of the monolayer did not show gross differences in

expression levels of any of the two catenins at different junctions within the cell monolayer. Thus, we did not find evidence that plakoglobin or β -catenin distribution, even after inflammatory stimulation would show local differences or point towards sites that might be related to preferred “hotspot” of diapedesis. This is different from preferential sites of ICAM-1 upregulation in EC monolayers which do indeed correlate with hotspots of diapedesis, as was shown by the van Buul lab.

2. Although plakoglobin knockdown markedly reduces leukocyte diapedesis, the mechanistic link between VE-cadherin tension modulation and leukocyte transendothelial migration (TEM) merits further clarification. Does plakoglobin-mediated tension facilitate the dissociation of VE-cadherin dimers or promote local internalization of VE-cadherin complexes? Moreover, how might these processes influence endothelial permeability during TEM? Given prior evidence from this group and others indicating that leukocyte TEM and vascular permeability can be uncoupled, additional discussion on this point would strengthen the manuscript.

Indeed, we speculate that the increase of mechanical force will challenge the trans-interaction between VE-cadherin molecules from adjacent cells. This will lead on average to a higher percentage of cadherin molecules which are disengaged and therefore available for endocytosis. We discussed this in the third paragraph of the discussion section.

As suggested by the reviewer, we have now also discussed (page 17) how this process of tension regulation that addresses VE-cadherin, could occur while at the same time there is increased tension on F-actin structures that surround transmigrating neutrophils and keep the emigration pore tight, as has been shown by Heemskerk et al (2016).

3. In addressing the in vivo tension measurements, the authors note the limitation in assessing neutrophil transmigration-induced VE-cadherin tension. It may be pertinent to reference prior phospho-myosin light chain (phospho-MLC) staining studies in inflamed cremaster muscle tissue (Heemskerk et al., 2016, Figure 6F), which provide evidence of actomyosin-generated tension during diapedesis. Incorporating this information would contextualize and reinforce the mechanistic narrative.

We agree with this comment and have now cited these results in the discussion, page 19.

Collectively, these data denote an unappreciated specificity in the molecular machinery underlying endothelial barrier regulation, demonstrating that plakoglobin, but not β -catenin, is required for leukocyte diapedesis, inflammatory induction of vascular permeability, and the transmission of mechanical forces across VE-cadherin. The study thus highlights plakoglobin as a critical determinant of endothelial function during inflammatory responses, with potential implications for targeting vascular dysfunction in pathological inflammation.

Dear Dietmar,

Thank you for submitting the revised version of your manuscript to The EMBO Journal. The study has now been seen by all original referees, who find that their main concerns have been addressed satisfactorily and recommend acceptance of the manuscript.

There now remain only a few editorial and formatting points that need to be addressed before I can extend official acceptance of the manuscript:

1. As part of the EMBO Press transparent editorial process, The EMBO Journal will publish online a Peer Review File to accompany accepted manuscripts. This file will be published in conjunction with your paper and will include the anonymous referee reports, your point-by-point response and all pertinent correspondence relating to the manuscript, including decision letters. Please note that the Author Checklist will be published at the end of the Peer Review File. Please let us know if you want to remove or not any figures or data from the Peer Review File prior to publication. Please note that retaining unpublished data in the Peer Review File means that these count as published and that the Peer Review File would need to be referenced in future publications.
2. There is an author name discrepancy between the manuscript and our online system: Jessica Heinrich (manuscript text) vs Jessica Heinrichs (online), please check.
3. CRedit has replaced the traditional author contributions section because it offers a systematic, machine-readable author contributions format that allows for more effective research assessment. Please remove the Authors Contributions from the manuscript and use the free text boxes beneath each contributing author's name in our online submission system to add specific details on the author's contribution. More information is available in our guide to authors.
4. Please rename "Conflict of interest" section into "Disclosure and competing interests statement" (further info: <https://www.embopress.org/page/journal/14602075/authorguide#conflictsofinterest>).
5. Figure panels for Fig 6B-C, Fig 10B-E and individual EV figure panels are not mentioned in the manuscript text. Please add the corresponding callouts.
6. We require a Data Availability Section at the end of Materials and Methods. As far as I can see, no data deposition in external databases is needed for this paper. If I am correct, then please state in this section: "This study includes no data deposited in external repositories". Further information can be found at <https://link.springer.com/partners/embo-press/editorial-policies#Data%20availability%20statement>.
7. During our routine image integrity checks, we observed that the blot images within the figure set appear pixelated under analysis. This is often a result of converting original 16-bit TIFF files to RGB format for publication. While this is not inherently problematic, it can give the impression of image alteration to critical readers. To address this, please upload blot images at the original resolution at which they were captured. Please also supply the blot source data at the same resolution. This will enable us to confirm the integrity of the complete figure set and enhance transparency for readers.
8. In our standard source data check, we have noted numerical repetitions in the source data for several figure panels. I have attached the corresponding files with the detected duplications labelled in colour. Please check and correct if needed. I appreciate that this could be caused by the specific measurement approaches or the calculations used. A brief explanation would be very helpful.
9. Our data editors have indicated that the exact p values should be provided in the legends of figures 1B, C; 2C, D, E; 3D, 4A, B; 5C, H; 7B, C; 8A, B 10 D, E; 11B, EV4.
10. Papers published in The EMBO Journal are accompanied online by a 'Synopsis' to enhance discoverability of the manuscript. It consists of A) a short (1-2 sentences) summary of the findings and their significance, B) 3-4 bullet points highlighting key results and C) a synopsis image that is 550x300-600 pixels large (width x height, jpeg or png format). You can either show a model or key data in the synopsis image. Please note that the image size is rather small and that text needs to be readable at the final size. Please send us this information together with the revised manuscript.

Thank you for giving us the chance to consider your manuscript for The EMBO Journal. I look forward to receiving the final version.

With best wishes,

Ieva

Ieva Gailite, PhD
Senior Scientific Editor
The EMBO Journal

Meyerhofstrasse 1
D-69117 Heidelberg
Tel: +4962218891309
i.gailite@embojournal.org

We realize that it is difficult to revise to a specific deadline. In the interest of protecting the conceptual advance provided by the work, we recommend a revision within 3 months (22nd Apr 2026). Please discuss the revision progress ahead of this time with the editor if you require more time to complete the revisions.

Referee #1:

This is a very interesting manuscript which shows the specificity of beta-catenin vs gamma-catenin (plakoglobin) at endothelial cell-cell adhesions. I applaud the authors for the revised manuscript which fully addressed my limited prior concerns. I believe the article will be of broad interest and is suitable for publication.

Referee #2:

In their re-submission, the authors have addressed key aspects of the concerns raised, most notably providing additional data on potential effect of the compensatory over-expression of B-catenin. However, it is surprising that no attempt was made to assess the protein level of B-catenin in venules of the KD model but this lack of data does not impact the overall impact of the study. Overall, whilst the revised work has not developed the molecular basis of the difference seen between the two catenins under investigation, the MS offers a significant insight into their differing biological functions.

Referee #3:

The authors have commented on all my concerns, and I am fully satisfied with the rebuttal, also their comments on the concerns of the other 2 reviewers.

The authors addressed the remaining editorial issues.

Dear Dietmar,

Thank you for addressing the final editorial requests. I am now pleased to inform you that your manuscript has been accepted for publication - congratulations!

Before we forward your manuscript to our publishers, we would like to propose some textual edits in the manuscript title, abstract and synopsis, which you can find in the attached text file. I have also selected a short blurb that will accompany the title of your manuscript in our online table of contents. Please take a look and let me know if any corrections or adjustments are needed.

You may qualify for financial assistance for your publication charges - either via a Springer Nature fully open access agreement or an EMBO initiative. Check your eligibility: <https://link.springer.com/journal/44318/how-to-publish-with-us>

If you have any questions, please do not hesitate to contact the Editorial Office. Thank you for this contribution to The EMBO Journal and congratulations on a nice study!

With best wishes,

leva

leva Gailite, PhD
Senior Scientific Editor
The EMBO Journal
Meyerhofstrasse 1
D-69117 Heidelberg
Tel: +4962218891309
i.gailite@embojournal.org